Host-Microbe Biology

# Single-Inclusion Kinetics of *Chlamydia trachomatis* Development

Travis J. Chiarelli,[a] Nicole A. Grieshaber,[a] Anders Omsland,[b] Christopher H. Remien,[a] Scott S. Grieshaber[a]

[a]Department of Biological Sciences, University of Idaho, Moscow, Idaho, USA
[b]Paul G. Allen School for Global Animal Health, College of Veterinary Medicine, Washington State University, Pullman, Washington, USA

**ABSTRACT** The obligate intracellular bacterial pathogen *Chlamydia trachomatis* is reliant on a developmental cycle consisting of two cell forms, termed the elementary body (EB) and the reticulate body (RB). The EB is infectious and utilizes a type III secretion system and preformed effector proteins during invasion, but it does not replicate. The RB replicates in the host cell but is noninfectious. This developmental cycle is central to chlamydial pathogenesis. In this study, we developed mathematical models of the developmental cycle that account for potential factors influencing RB-to-EB cell type switching during infection. Our models predicted that two categories of regulatory signals for RB-to-EB development could be differentiated experimentally, an "intrinsic" cell-autonomous program inherent to each RB and an "extrinsic" environmental signal to which RBs respond. To experimentally differentiate between mechanisms, we tracked the expression of *C. trachomatis* development-specific promoters in individual inclusions using fluorescent reporters and live-cell imaging. These experiments indicated that EB production was not influenced by increased multiplicity of infection or by superinfection, suggesting the cycle follows an intrinsic program that is not directly controlled by environmental factors. Additionally, live-cell imaging revealed that EB development is a multistep process linked to RB growth rate and cell division. The formation of EBs followed a progression with expression from the *euo* and *ihtA* promoters evident in RBs, while expression from the promoter for *hctA* was apparent in early EBs/IBs. Finally, expression from the promoters for the true late genes, *hctB*, *scc2*, and *tarp*, was evident in the maturing EB.

**IMPORTANCE** *Chlamydia trachomatis* is an obligate intracellular bacterium that can cause trachoma, cervicitis, urethritis, salpingitis, and pelvic inflammatory disease. To establish infection in host cells, *Chlamydia* must complete a multiple-cell-type developmental cycle. The developmental cycle consists of specialized cells, the EB cell, which mediates infection of new host cells, and the RB cell, which replicates and eventually produces more EB cells to mediate the next round of infection. By developing and testing mathematical models to discriminate between two competing hypotheses for the nature of the signal controlling RB-to-EB cell type switching, we demonstrate that RB-to-EB development follows a cell-autonomous program that does not respond to environmental cues. Additionally, we show that RB-to-EB development is a function of chlamydial growth and division. This study serves to further our understanding of the chlamydial developmental cycle that is central to the bacterium's pathogenesis.

**KEYWORDS** bacterial development, chlamydia, live-cell imaging, mathematical modeling, infectious disease

Chlamydiae are bacterial pathogens responsible for a wide range of diseases in both animal and human hosts (1). *Chlamydia trachomatis*, a human-adapted pathogen, comprises over 15 distinct serovars causing both trachoma, the leading cause of

Address correspondence to Scott S. Grieshaber, scottg@uidaho.edu.

Live cell imaging reveals chlamydial developmental kinetics

preventable blindness, and sexually acquired infections (2). According to the CDC, *C. trachomatis* is the most frequently reported sexually transmitted infection in the United States, costing the American health care system nearly $2.4 billion annually (3, 4). These infections are widespread among all age groups and ethnic demographics, infecting ~3% of the human population worldwide (5). In women, untreated genital infections can result in pelvic inflammatory disease, ectopic pregnancy, and infertility (6–8). Every year, there are over 4 million new cases of *C. trachomatis* sexually transmitted infections in the United States (6, 9) and an estimated 92 million cases worldwide (10).

*Chlamydia*-related disease is entirely dependent on the establishment and maintenance of the pathogen's unique intracellular niche, the chlamydial inclusion, where the bacteria replicate and undergo a biphasic developmental cycle. This cycle generates two unique developmental cell forms: the elementary body (EB) and the reticulate body (RB). The EB cell type mediates host cell invasion via pathogen-mediated endocytosis, while the RB cell type is replication competent but cannot initiate host cell infection (11). For *C. trachomatis* serovar L2, the cycle begins when the EB binds to a host cell and initiates uptake through the secretion of effector proteins by a type III secretion system (12). During entry, the EB is engulfed by the host cell plasma membrane, forming the inclusion vacuole that is actively modified by *Chlamydia* to block interaction with the host endocytic/lysosomal pathway (13). The inclusion continues to mature as the EB cell form transitions to the RB cell form. The time from host cell contact to the formation of the mature inclusion containing replication-competent RBs is ~11 h (14). The formation of infectious EB cells occurs reliably between 18 and 20 h postinfection (hpi) (15). Regulatory control of the transition between the RB and EB is critical for the chlamydial life cycle, as *Chlamydia* must balance replication versus production of infectious progeny. How *Chlamydia* regulates this process is currently unclear, although there have been multiple hypotheses proposed to explain the control of the developmental cycle. Regulatory mechanisms, such as RB access to or competition for inclusion membrane contact (16), reduction in RB size (14), or responses to changes in nutrient availability (17), all have been proposed to control or influence RB-to-EB cell switching.

In this study, we used mathematical modeling to guide experiments to distinguish between factors that influence RB-to-EB development. The chlamydial life cycle was modeled using systems of differential equations. Each model was tested under simulated conditions that indicated that extrinsic versus intrinsic control of EB development could be distinguished experimentally. To test the model predictions, a live-cell imaging system in combination with promoter-reporter constructs was developed to monitor the developmental cycle in real time at the single-inclusion level. We show that neither the limiting membrane hypothesis nor the intrainclusion nutrient-limiting hypothesis are consistent with our experimental results and that EB development likely follows a cell-autonomous program. Additionally, we show that this intrinsic program is dependent on RB growth and cell division.

## RESULTS

**Modeling chlamydial development.** We developed two mathematical models that represent potential driving forces in promoting EB development. Each model is a system of ordinary differential equations (ODE) that tracks RBs, intermediate bodies (IBs), and EBs over time (Fig. 1; see also Fig. S1 in the supplemental material). In these models, the development of the EB is controlled by an inhibitory signal that is intrinsic to each bacterium or is environmental, i.e., shared between the bacteria (Fig. 1A and B). The nature of the signal was not specified beyond an inhibitory effect on EB production at high concentrations and its consumption by RBs. The regulatory nature of this signal could be either positive, as in quorum sensing, or negative, such as nutrient limitation. For our simplified model system, we implemented a negative regulator, but the model will generate identical outputs if the regulator is positive in nature. For each of the two models, the signal is consumed by the bacteria over time, and, once depleted, RB-to-EB conversion commences. The models differ in whether all the RBs in the inclusion compete for one pool of this signal or whether each RB contains an independent

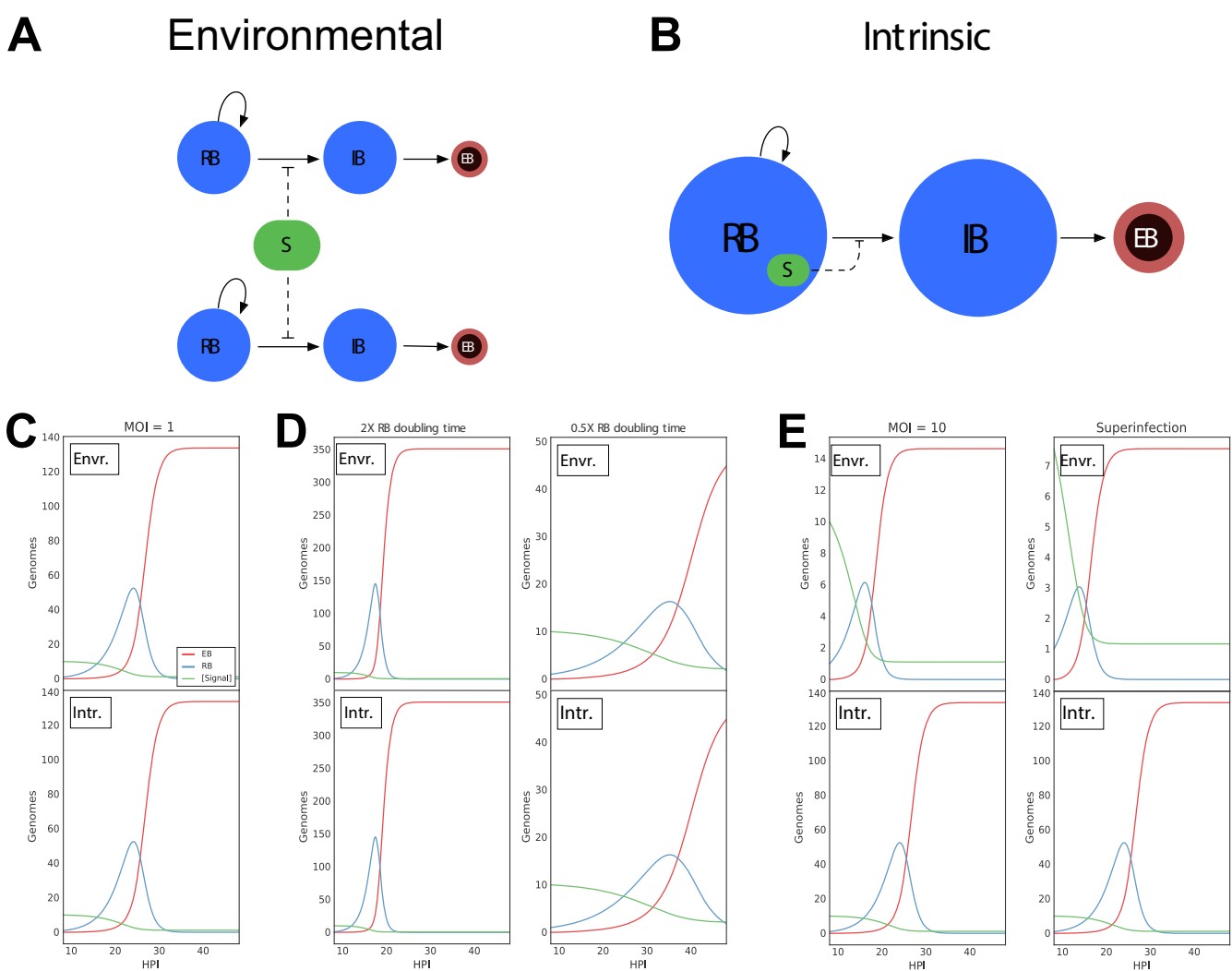

**FIG 1** Schematic and simulations of environmental and intrinsic models. Both models assume that the mechanism of RB/EB conversion is in response to signal concentration. High signal concentration prevents RB/EB conversion, and RB replication continues. As RBs replicate, the signal is consumed. Once the signal is depleted past a given threshold, RBs convert to IBs, which then convert to EBs. (A) Schematic of the environmental signal model. The RBs compete for a single pool of signal (S). (B) Schematic of the intrinsic model. Each RB contains its own signal, eliminating competition between RBs. (C) Simulations of the two models (environmental and intrinsic) using a multiplicity of infection (MOI) of 1 and an RB generation time of 2.27 h produced results that mimic the general kinetics of the chlamydial cycle and were indistinguishable from each other. (D) Simulations of RB doubling times of 1.13 h (half the measured RB doubling time) resulted in a reduced time to EB production, whereas 4.54 h ($2\times$ the measured RB doubling time) increased time to EB production. However, both models (environmental and intrinsic) produced the same outcome. (E) Simulations using an MOI of 10 predicted EB conversion to occur more rapidly in the environmental signal model but to remain unchanged in the intrinsic model. Similarly, simulations of the models using a time-delayed superinfection resulted in RB-to-EB conversion occurring more rapidly in the environmental model but remaining unchanged in the intrinsic model.

internal pool of the inhibitory signal. The output of both models mimics the general kinetics of the chlamydial developmental cycle. Both models produced identical outputs when a multiplicity of infection (MOI) of 1 was simulated (Fig. 1C). When a change in the replication rate of *Chlamydia* was simulated, the two models again responded similarly, showing that an increased replication rate led to earlier EB production, while a decreased replication rate resulted in delayed EB production (Fig. 1D). However, the models produced dramatic kinetic differences with a simulated increase in MOI or time-delayed superinfection. Both simulated conditions caused EB formation to occur sooner in the environment-based signal model but had no effect on EB production when modeled with an intrinsic signal (Fig. 1E). These data indicate that it is possible to experimentally differentiate between whether an environmental signal or an intrinsic program triggers EB development.

**Development of a live-cell reporter system to monitor the chlamydial developmental cycle.** To experimentally differentiate between mechanisms of differentia-

tion based on the response to an environmental or intrinsic signal, we developed a live-cell imaging system using promoter constructs to monitor the chlamydial developmental cycle. The reporter constructs were designed using the promoters of chlamydial genes that are differentially regulated between the RB and EB forms (18). To generate an RB reporter, the promoter of *ihtA* was used to drive enhanced green fluorescent protein (EGFP) expression. The sRNA IhtA is expressed early upon infection and negatively regulates the EB-specific gene *hctA* (19). To generate an EB reporter, the promoter and first 30 nucleotides (nt) of the late gene *hctA* were used to drive the expression of the GFP variant Clover. HctA is a small histone-like protein that is involved in the condensation of the chlamydial genome to form the compact nucleoid characteristic of the EB (20). The upstream promoter region as well as the first 10 codons of the open reading frame (ORF) of *hctA* were used to construct this reporter, as the regulation of HctA expression involves both the promoter and the IhtA binding site contained in the beginning of the ORF (21). Each reporter was transformed into *C. trachomatis*, generating the strains *Ctr-ihtA*prom-EGFP and *Ctr-hctA*prom-Clover (see Table S1 in the supplemental material). The chlamydial transformants were used to track the developmental cycle of each strain using live-cell time-lapse microscopy and particle tracking to quantify the fluorescent expression of individual inclusions over time (22). This technique allows for the tracking of gene expression in multiple individual inclusions over the entire developmental cycle while avoiding the inherent variability of whole-population studies on an asynchronous infection. A detailed description of the system is described in our recently published paper (23). To verify that the fluorescent reporters accurately reflected the developmental cycle, total chlamydial growth was determined by measuring genomic copies by quantitative PCR (qPCR) and EB production by a replating assay to quantify inclusion-forming units (IFU). EGFP expression from the *ihtA* promoter was first detected at ~10 hpi and started to level off at ~28 hpi (Fig. 2A). The initial expression from the *ihtA* promoter was in good agreement with the initiation of RB genomic replication, as demonstrated by genome copies (Fig. 2A). The initiation of RB replication signals the end of the EB-to-RB transition after cell entry. Imaging of the *hctA* promoter-reporter revealed that the Clover signal could be detected first at ~18 hpi (Fig. 2B). Again, these data were in good agreement with the production of infectious progeny, as EBs were first detected at ~20 hpi (Fig. 2B). We measured >50 individual inclusions per strain and found very little interinclusion variability in the timing of the initiation of expression (Fig. 2C and D). This uniformity in developmental timing can be appreciated in a live-cell time-lapse movie of *Ctr-hctA*prom-Clover infections (Movie S1). The close agreement between classic methods for monitoring the chlamydial developmental cycle (IFU and genome copies) and the single-inclusion-based fluorescent reporter system described here demonstrates that this system accurately reflects the developmental cycle.

**Chlamydial development is growth rate dependent.** Both models predicted that changes in growth rate would be reflected in EB production kinetics (Fig. 1D). There is generally a linear relationship between temperature and the square root of growth rate in bacteria (24). Therefore, to validate the predictions of our two models, we monitored *Ctr-ihtA*prom-EGFP and *Ctr-hctA*prom-Clover at three temperatures, 35°C, 37°C (control), and 40°C. As expected, at the lower temperature of 35°C, the EB-to-RB lag time increased dramatically and *ihtA*prom-EGFP expression increased more slowly than that of the 37°C control (Fig. 3A). The lower replication rate at 35°C was also reflected in measured genome copies (Fig. 3B). Conversely, the lag time to fluorescence detection was reduced and fluorescence increased faster than the control when grown at 40°C (Fig. 3A). As predicted by our models, time to EB production was also shifted by changes in growth rate, as *hctA*prom-Clover expression began earlier at 40°C and was delayed at 35°C (Fig. 3C). These results were verified by measuring the production of infectious progeny (Fig. 3D) and are consistent with previously published literature where *Chlamydia* growth at 33°C was slowed in both inclusion and EB development (25). Taken together, these data provide strong evidence that the cycle is growth rate

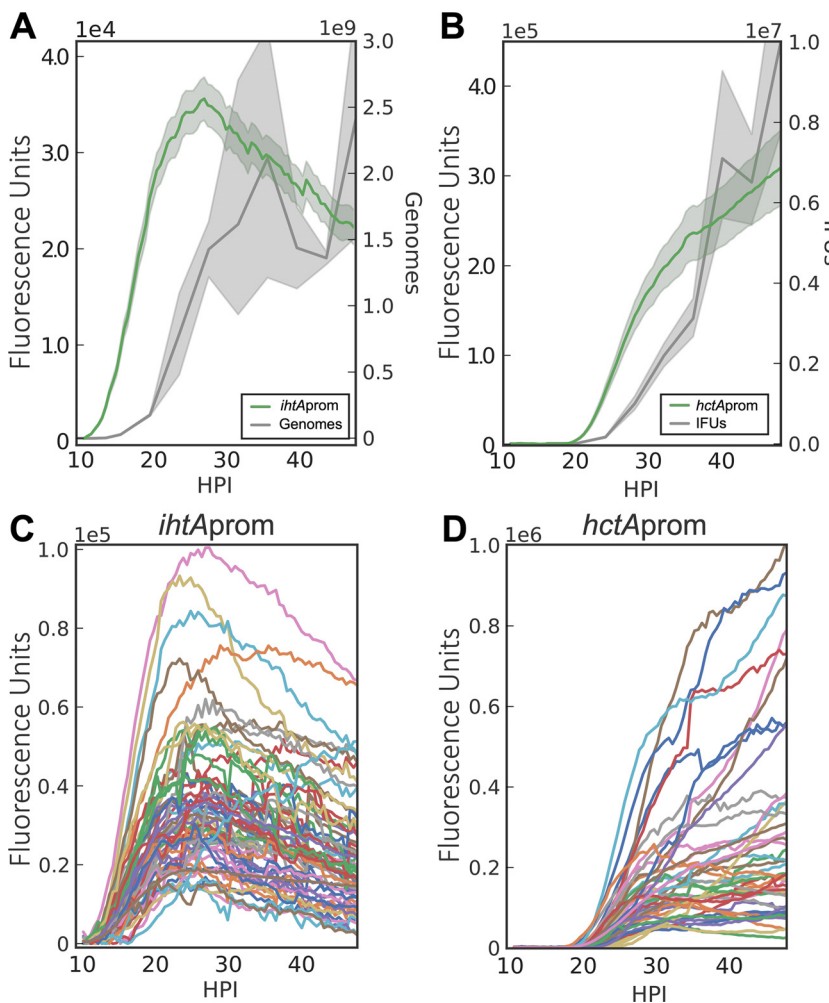

**FIG 2** Live-cell fluorescent imaging of chlamydial development. Cell type-specific fluorescent reporters were created to track chlamydial development in real time. Infections with purified *Ctr*-L2-prom EBs were synchronized and fluorescence microscopy, and qPCR/reinfection assays were run simultaneously. (A and B) The averages of *ihtA*prom-EGFP and *hctA*prom-Clover expression intensities from >50 individual inclusions monitored via automated live-cell fluorescence microscopy throughout the developmental cycle compared to genome copies and IFU, respectively. (C and D) The fluorescence intensities of >50 individual inclusions tracked via live-cell microscopy throughout the developmental cycle. The fluorescent unit cloud represents standard error of the mean (SEM) genome copies, and the IFU cloud represents 95% confidence intervals (CI). *y* axes are denoted in scientific notation.

dependent and that our experimental system accurately detected changes in chlamydial development.

**EB development is controlled by intrinsic factors and not environmental factors.** The two mathematical models differ principally in the source of the EB development signal: internal versus environmental. The models produced divergent outcomes under conditions where bacteria are competing for a host cell or an intrainclusion signal versus a signal internal to each RB. Simulations predicted that the time to EB production would be measurably affected by increasing the MOI if the signal was environmental (competitively consumed) but would be unchanged if the signal was intrinsic (internal to each RB) (Fig. 1E). To more accurately assay EB development by live-cell imaging, two additional EB gene reporters were constructed. The promoters and first 30 nt of *hctB* and *scc2* were inserted upstream of Clover and transformed into *C. trachomatis*, creating *Ctr-hctB*prom-Clover and *Ctr-scc2*prom-Clover, respectively. Like HctA, HctB is a small histone-like protein that is involved in EB nucleoid formation (26), while Scc2 is a chaperone for type III secretion effector proteins (27). Our published

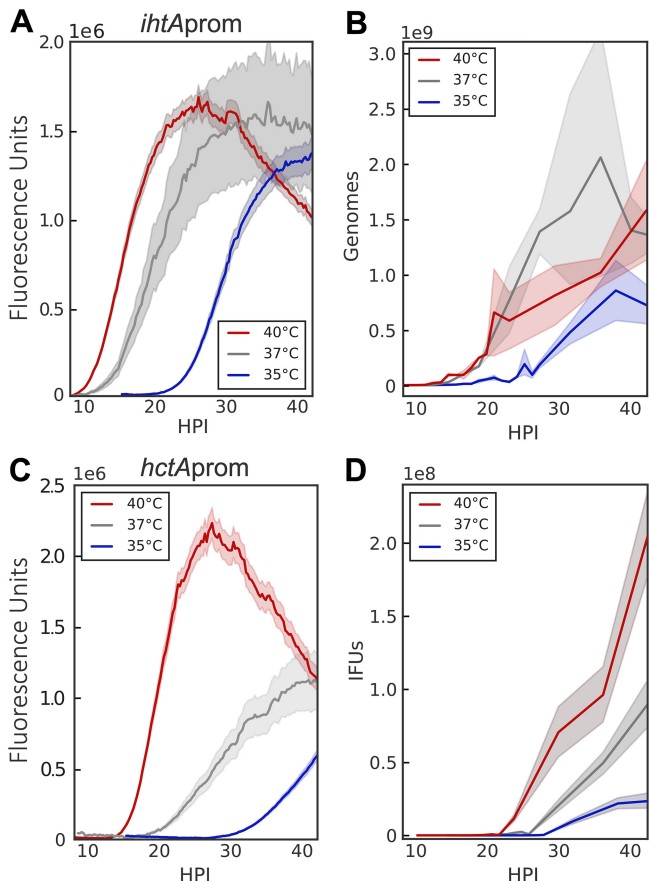

**FIG 3** RB replication and EB conversion are growth rate dependent. The ability of the promoter-reporter system to monitor differences in RB replication and EB conversion was tested by altering the growth temperature (35°C, blue; 37°C, gray; 40°C, red). (A) The averages of *ihtA*prom-EGFP expression intensities of >50 individual inclusions monitored from 9 to 42 hpi via live-cell fluorescence microscopy. (B) Genome copies were measured between 2 and 42 hpi by qPCR. (C) The averages of *hctA*prom-Clover expression intensities of >50 individual inclusions monitored from 9 to 42 hpi via live-cell fluorescence microscopy. (D) EB conversion (IFU) was quantified via replating assay from 11 to 42 hpi. The fluorescent unit cloud represents standard error of the mean (SEM) genome copies, and the IFU cloud represents 95% confidence intervals (CI). *y* axes are denoted in scientific notation.

transcriptome sequencing (RNA-seq) data showed that the transcripts for *hctB* and *scc2* were expressed late, corresponding to the timing of EB production (18). Monolayers were infected with each of the four strains, with MOIs ranging from 1 to 32 infectious EBs per host cell, and imaged every 30 min for 40 h. The MOI was calculated by infection with a 2-fold dilution series and back calculating from an observed MOI of 1. The fluorescent signals were normalized by MOI, as this more closely represents fluorescence per RB. Expression initiation of the RB reporter *ihtA*prom, and the EB reporters *hctA*prom, *hctB*prom, and *scc2*prom, did not vary as a function of MOI (Fig. 4A to D). The lack of MOI response for the expression of EB genes corresponded closely with EB production as measured by a reinfection assay (Fig. 4E). Of note is the dramatic difference in the timing of expression between the late genes. *hctA*prom-Clover expression was initiated at ~18 h postinfection, while *hctB*prom-Clover and *scc2*prom-Clover expression was initiated ~3 h later at ~21 hpi.

Our models predicted that both MOI and superinfection would aid in differentiating between cell-autonomous and environmentally influenced development (Fig. 1E). The MOI data suggested that RB-to-EB developmental switching is not influenced by the host intracellular or the intrainclusion environment but rather is triggered by a signal intrinsic to *C. trachomatis*. To further differentiate between these possibilities, we measured RB and EB gene expression under superinfection

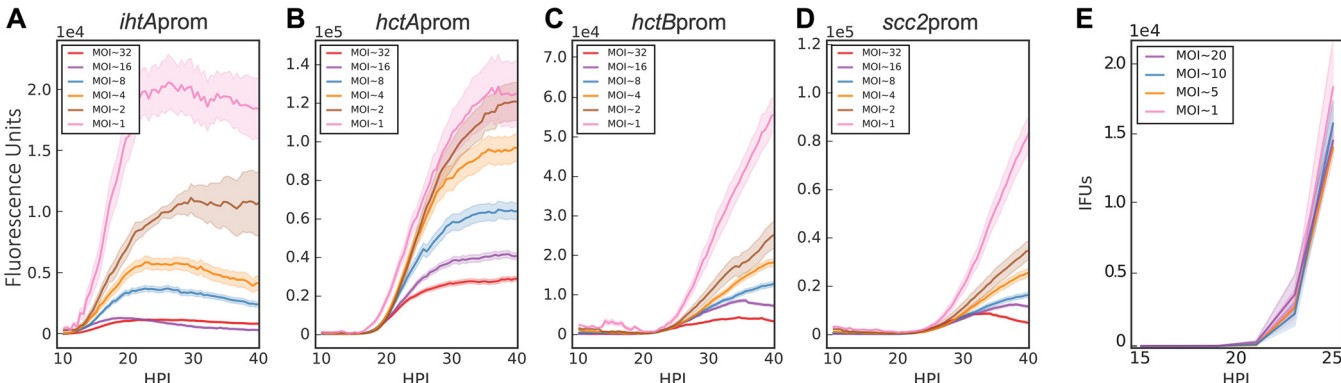

**FIG 4** MOI does not affect initiation of RB-to-EB conversion. Host cells were infected with purified *Ctr*-L2-prom EBs at an MOI of 1 to 32. (A to D) Averages of *ihtA*prom-EGFP, *hctA*prom-Clover, *hctB*prom-Clover, and *scc2*prom-Clover expression intensities from >50 individual inclusions monitored via automated live-cell fluorescence microscopy throughout the developmental cycle. Fluorescent intensities were normalized by the respective MOI. (E) EB development (IFU) was measured at MOIs from 1 to 20 and was quantified via a replating assay. EBs were harvested at 2-h intervals from 15 to 25 hpi. IFU data were normalized by the respective MOI. The fluorescent unit cloud represents standard error of the mean (SEM) genome copies, and the IFU cloud represents 95% confidence intervals (CI). *y* axes are denoted in scientific notation.

conditions. The chlamydial inclusion is derived from the plasma membrane, and interaction with the endocytic membrane system is actively blocked by *Chlamydia* (13). When multiple EBs infect a cell, they each create individual inclusions that traffic to the microtubule-organizing center (MTOC) of the host cell (28). This trafficking, along with the expression of IncA, a protein that promotes fusion of individual inclusions, culminates in homotypic inclusion fusion, resulting in a single chlamydial inclusion per host cell (29, 30). Our environmental signal model predicted that the developmental cycle of *Chlamydia* under superinfection conditions would be dramatically altered (decreased time to EB production) as a function of the developmental stage of the first infection. To test this, cells were infected with unlabeled *C. trachomatis* L2 for 6, 12, and 18 h prior to a second infection with the indicated *C. trachomatis* L2 reporter strains and imaged starting at 9 h after secondary infection (Fig. 5). Fluorescent signals were measured for inclusions that were verified to be superinfected by imaging for both differential interference contrast (DIC) and fluorescence, i.e., inclusions containing both labeled and unlabeled *Chlamydia* (Fig. 5A). Superinfection at any time after initial infection had no effect on the initiation of expression of either *ihtA*prom-EGFP or *hctA*prom-Clover (Fig. 5B and C). The lack of effect on late gene expression was verified with two other late promoter-reporter strains, *Ctr-hctB*prom-Clover and *Ctr-Scc2*prom-Clover, 12 h postsuperinfection (Fig. 5D and E). We verified that superinfection had no effect on the initial production of infectious progeny by performing a replating assay in the presence of spectinomycin (Fig. 5F).

To further examine any effect of the intrainclusion environment versus the host intracellular environment, we took advantage of a *Chlamydia* mutant that does not express IncA and, therefore, is defective in homotypic inclusion fusion (29). Cells were preinfected with an isogenic mutant pair, either *C. trachomatis* J (*incA* positive and fusogenic [31]) or *C. trachomatis* Js (*incA* negative and nonfusogenic [31]) for 18 h, and then were superinfected with *Ctr-ihtA*prom-EGFP or *Ctr-hctA*prom-Clover and imaged starting at 9 h postsuperinfection (Fig. 5G). Again, there was no apparent change in kinetics between infection alone (no superinfection), superinfection with inclusion fusion, or superinfection without fusion (Fig. 5H and I). Taken together, these data suggest that the timing of RB-to-EB development is an intrinsic preprogrammed property of *Chlamydia* and does not respond to environmental signals.

**Chlamydial cell division is required for EB development.** Time to EB development responded to RB growth rate, suggesting that chlamydial cell division is critical for development (Fig. 3). To test the role of cell division in EB development, RB replication was halted by treating infected cells with penicillin G (Pen). *C. trachomatis* does not use

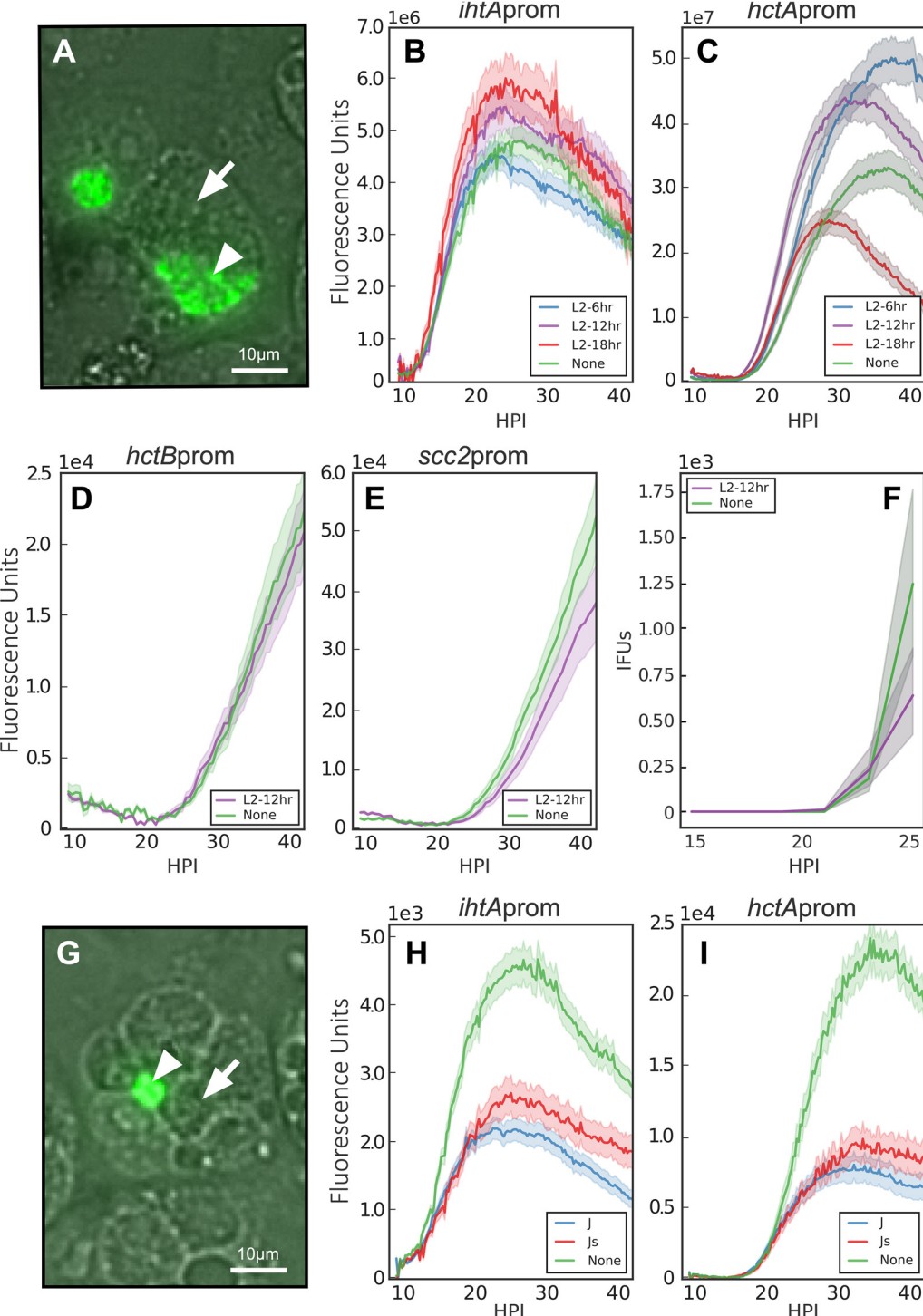

**FIG 5** Superinfection does not affect RB-to-EB conversion. Host cells were infected with nonfluorescent *C. trachomatis* EBs followed by a secondary infection with *Ctr*-L2-prom EBs at 6, 12, or 18 hpi, and the fluorescent output was compared to that of cells that had not been infected with a primary infection (none). Infections were imaged starting at 9 h postinfection with the *Ctr*-L2-prom strains. (A) Live-cell fluorescence/DIC image of 18-h L2 superinfection with *Ctr-hctA*prom-Clover at ×20 magnification (30 h after *Ctr-hctA*prom-Clover infection). Fluorescent signals were measured in inclusions containing both GFP-expressing *C. trachomatis* (arrowhead) and nonfluorescent *C. trachomatis* (arrow). Scale bar, 10 μm. (B and C) The averages of *ihtA*prom-EGFP and *hctA*prom-Clover expression intensities from >50 individual inclusions monitored via automated live-cell fluorescence microscopy during no superinfection (none) and 6, 12, and 18 h *C. trachomatis* L2 superinfections. (D and E) The average fluorescent intensities of >50 individual inclusions using *Ctr-hctB*prom-Clover or *Ctr-scc2*prom-Clover measured with no superinfection (none) or 12 h *C. trachomatis* L2 superinfection. (F) EBs were harvested at 2-h intervals from 15 to 25 h after *Ctr*-L2-prom infection and quantified by replating assay. (G) Live-cell fluorescence/DIC image of cells infected with *C. trachomatis* Js and superinfected with *Ctr-hctA*prom-Clover. The image was

peptidoglycan as a structural sacculus and does not contain a peptidoglycan cell wall. Instead, peptidoglycan aids cell septation by forming a ring at the cleavage furrow (32). Therefore, Pen treatment blocks cell septation but not cell growth.

To assess the effects of Pen treatment on chlamydial developmental kinetics, an additional early gene promoter-reporter, *euo*prom-Clover, was constructed. EUO (early upstream ORF) is a transcriptional repressor that selectively regulates promoters of *C. trachomatis* late genes and was highly expressed in our RNA-seq data set (18, 33). Cells infected with *Ctr-euo*prom-Clover or *Ctr-hctA*prom-Clover were treated with Pen at 14 hpi and imaged for a further 34 h (Fig. 6). The *euo*prom-Clover signal after Pen treatment continued to increase, as did the size of the aberrant RB cells (Fig. 6B and 7A and B). The expression of *euo*prom-Clover in the presence of Pen also matched the increase in genome copies, which, as previously reported (34), was also Pen insensitive (Fig. 6D). Unlike *euo*prom-Clover expression, the *hctA*prom-Clover signal was dramatically affected by Pen treatment (Fig. 6B). The expression of *hctA*prom-Clover was initially repressed by Pen treatment at 14 hpi compared to that of untreated samples; however, expression was initiated ~9 h after treatment. We explored this late gene expression behavior further using three other late gene promoter strains, *Ctr-hctB*prom-Clover, *Ctr-scc2*prom-Clover, and *Ctr-tarp*prom-Clover (Fig. 6B and Fig. S2). The Clover expression patterns driven by *hctB*prom, *scc2*prom, and *tarp*prom were dramatically different from that of *hctA*prom, as none showed Clover expression in the Pen-treated samples (Fig. 6B and Fig. S2). The lack of *hctB*prom, *scc2*prom, and *tarp*prom gene expression corresponded to the lack of production of infectious progeny during Pen treatment, suggesting that these genes can be considered true EB genes (Fig. 6E).

To further investigate the role of chlamydial cell division in EB development, we tested the effects of a second antibiotic that targets peptidoglycan synthesis, D-cycloserine (DCS). DCS is a cyclic analogue of D-alanine and inhibits peptidoglycan synthesis (35). Again, *euo*prom-Clover expression was measured over time after DCS treatment at 14 hpi. The kinetics of expression of *euo*prom-Clover was similar to that of Pen-treated and untreated samples (Fig. 6A to C). The expression kinetics of the late gene reporters after DCS treatment also mimicked Pen treatment. DCS-treated inclusions never expressed Clover from *hctB*prom or *scc2*prom reporters but did express from the *hctA*prom reporter with a similar ~9-h delay (Fig. 6C). Although the kinetics were similar to those of Pen treatment among all reporters, the aberrant RBs did not grow as large as those treated with Pen (Fig. 7).

Treatment with penicillin has been reported to induce aberrant RBs that continue to metabolize and increase in size but do not produce infectious progeny (36, 37). Pen, other antibiotic treatments, and nutrient limitation are all reported to induce a persistent state in *Chlamydia* (38). Therefore, we explored the effect of interferon gamma (IFN-γ)-induced persistence on cell-type-specific gene expression. While Pen and DCS induce persistence through their effects on peptidoglycan synthesis, IFN-γ causes an aberrant state by starving *Chlamydia* of tryptophan (39). HeLa cells were used as opposed to Cos7 cells, as the former responds to human IFN-γ (hIFN-γ). Cells were treated with IFN-γ 24 h prior to infection with the *Ctr-ihtA*prom-EGFP or *Ctr-hctA*prom-Clover strain. Imaging of these constructs showed that no signal was produced from either promoter construct (Fig. S3). We also treated cells with the iron chelator bipyridyl, which is reported to have regulatory overlap of tryptophan regulation in *Chlamydia* (40). Bipyridyl treatment also resulted in no signal produced from either promoter construct (Fig. S3).

**FIG 5** Legend (Continued)
taken 30 h after *Ctr-hctA*prom-Clover infection at ×20 magnification. Fluorescent signals were measured from inclusions in cells that contained both fluorescent *Ctr-hctA*prom-Clover (arrowhead) and unfused nonfluorescent *C. trachomatis* Js (arrow). Scale bar, 10 μm. (H and I) The average fluorescent intensity of >50 individual inclusions containing *ihtA*prom-EGFP and *hctA*prom-Clover measured with no superinfection (none), *C. trachomatis* J, or *C. trachomatis* Js superinfections. The fluorescent unit cloud represents standard error of the mean (SEM) genome copies, and the IFU cloud represents 95% confidence intervals (CI). *y* axes are denoted in scientific notation.

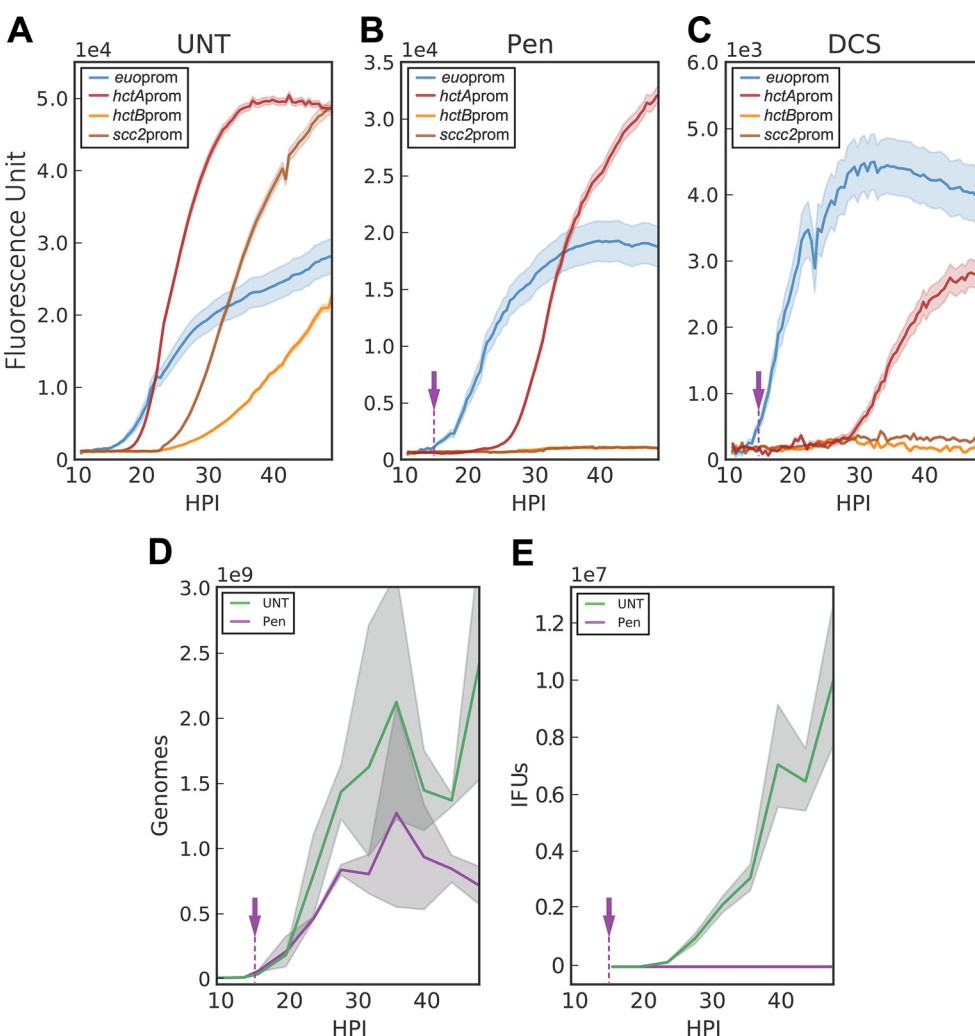

**FIG 6** Inhibition of chlamydial cell division inhibits EB conversion. Host cells were infected with purified *Ctr*-L2-prom EBs followed by treatment with penicillin G, D-cycloserine, or vehicle only at 14 hpi (purple arrow). (A to C) The averages of RB (*euo*prom-Clover), IB (*hctA*prom-Clover), and EB (*hctB*prom-Clover and *scc2*prom-Clover) expression intensities from >50 individual inclusions monitored via automated live-cell fluorescence microscopy in cells treated with vehicle only (UNT), penicillin (PEN), or D-cycloserine (DCS), respectively. (D) Quantification of genome copy numbers for vehicle only (UNT)- and penicillin (PEN)-treated cells measured using qPCR. (E) Quantification of EB development for vehicle only (UNT)- and penicillin (PEN)-treated cells via replating assay. EBs were harvested at 4-h intervals from 16 to 48 hpi. The fluorescent unit cloud represents standard error of the mean (SEM) genome copies, and the IFU cloud represents 95% confidence intervals (CI). *y* axes are denoted in scientific notation.

Data obtained from Pen- and DCS-treated infections support a role for cell division in chlamydial development. To further explore this observation, cells were treated with Pen every 2 h starting at 16 hpi. To visualize both RBs and EBs in the same inclusion during the developmental cycle, two dual promoter constructs were developed, creating *Ctr-hctA*prom-mKate2/*ihtA*prom-mNeonGreen and *Ctr-hctB*prom-mKate2/*euo*prom-Clover. Cells were infected with the dual promoter strains and imaged every 30 min starting at 14 h postinfection (Fig. 8). Expression levels of the fluorescent proteins driven by the early, early-late, and late promoters in response to Pen treatment were strikingly different. The *euo*prom signal increased compared to that of untreated infections almost immediately after Pen was added, regardless of the timing of treatment (Fig. 8A). This was also true for the other early promoter-reporter, *ihtA*prom (Fig. S4). Signal from the late promoter *hctB*prom was completely inhibited but only after a ~10-h delay, again regardless of when Pen was added (Fig. 8B). Conversely, *hctA*prom signal was inhibited very quickly after Pen treatment, but expression resumed after an ~9-h delay (Fig. 8C). Confocal

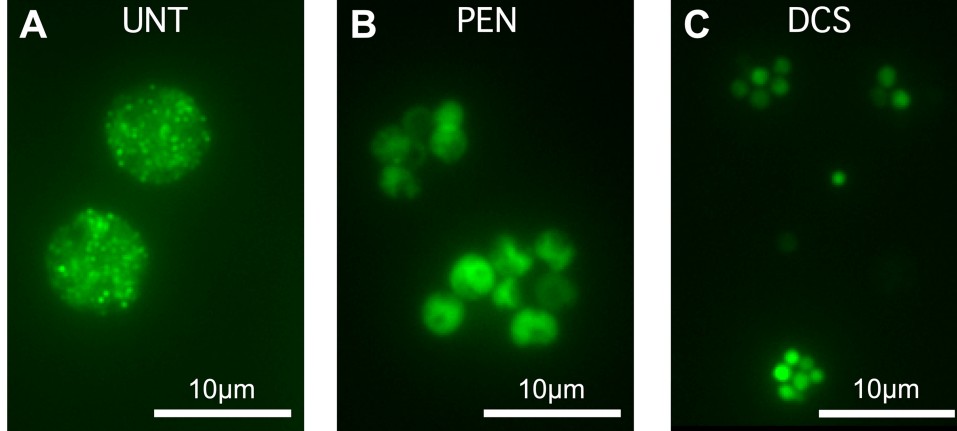

**FIG 7** Penicillin G and D-cycloserine induce aberrant cell forms. Host cells were infected with *Ctr-hctA*prom-Clover EBs followed by treatment with penicillin or D-cycloserine at 14 hpi. Live-cell fluorescence images were acquired at 40 hpi. (A) Untreated (UNT), vehicle only. (B) Penicillin (PEN) treated. (C) D-cycloserine (DCS) treated. Magnification, ×40. Scale bar, 10 μm.

images of Pen-treated cells indicated that *ihtA*prom-mNeonGreen and *euo*prom-Clover expression was evident only in the large aberrant cells (Fig. 9). However, there was a striking difference in cell type expression between the late promoters *hctA*prom and *hctB*prom. Like *ihtA*prom and *euo*prom, *hctA*prom-mKate2 expression was localized to large aberrant cells. In contrast, *hctB*prom-mKate2 expression was restricted to nonaberrant small cells that resembled EBs (Fig. 9).

**EB gene expression increases linearly until cell death.** Our data suggest that initial RB-to-EB development follows an intrinsic program and does not respond to environmental cues. However, the data show significant variability at >36 hpi. To better understand the kinetics of chlamydial development late during infection, well-separated individual inclusions were monitored from when fluorescence could first be detected until lysis of the inclusion or cell. The dual promoter strain, *Ctr-hctB*prom-mKate2/*euo*prom-Clover, was used to identify early inclusions and monitor late gene expression. Expression from each promoter in isolated individual inclusions was monitored for >65 hpi (Fig. 10A and Movie S2). Late in infection, gene expression from isolated inclusions differed significantly from aggregated expression data. *euo*prom-Clover expression in each individual inclusion followed a similar pattern, a lag phase and then a short exponential phase, followed by an expression plateau at ~24 hpi, which was maintained until cell death (Fig. 10A and Movie S2). *hctB*prom expression showed a short exponential growth phase followed by continuous linear gene expression ($R^2 = 0.99$) until cell lysis (Fig. 10A, graph 3, and Movie S2). Late in infection (>36 hpi), a subset of inclusions/cells lysed (Movie S2), which contributed to the increased signal variability through loss of fluorescence, resulting in aggregate gene expression data mimicking a stationary phase. The data from single inclusions suggest that the *Chlamydia* isolates are not responding to depleting resources of the host cell late in infection, as the slope is linear until lysis. Although growth is linear for every inclusion, the rate differs between inclusions in different cells (Fig. 10A, graph 2), suggesting that the growth rate of *Chlamydia* is set by a limiting nutrient inside the cell that is maintained at a steady state, producing a linear expression curve (Fig. 10A, graph 2, and Movie S2). Linear expression kinetics was also seen in cells grown at various temperatures. Infected cells grown at 35°C, 37°C, and 40°C all showed linear *hctB*prom expression, with slopes varying significantly with temperature, at 344, 499, and 713 fluorescence units/h, respectively (Fig. S5).

All data presented thus far were collected from infections in the presence of cycloheximide. Monolayers were treated with cycloheximide to block host cell division, which reduces cell migration and improves live-cell imaging. Cycloheximide is a

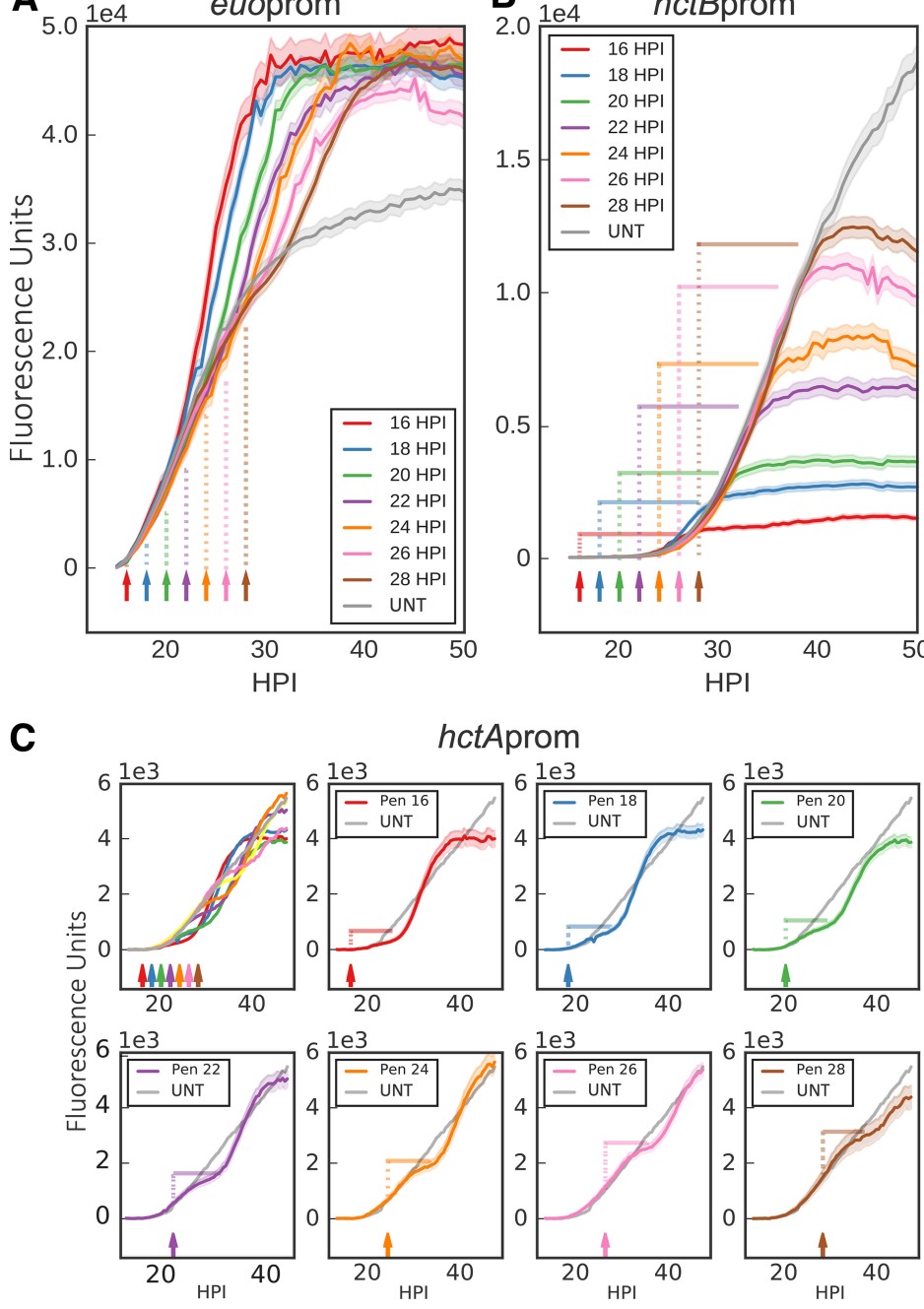

**FIG 8** Inhibiting chlamydial cell division inhibits further EB conversion. Host cells were infected with *Ctr*-L2-prom EBs followed by treatment with penicillin (Pen) at 2-h intervals starting at 16 hpi or without treatment (UNT). Arrows and vertical dotted lines indicate the addition of penicillin. (A) The averages of *euo*prom-Clover (RB) expression intensities from >50 individual inclusions monitored via automated live-cell fluorescence microscopy for each penicillin treatment (time series starting at 16 hpi) and no treatment (UNT). (B) The averages of *hctB*prom-mKate2 (EB) fluorescence from >50 individual inclusions. Horizontal solid lines indicate time to maximum expression. (C) The averages of *hctA*prom-mKate2 (IB) fluorescence from >50 individual inclusions. *hctA*prom-mKate2 graphs are separated for clarity. Horizontal solid lines indicate time to reinitiation of expression. The cloud represents SEM. *y* axes are denoted in scientific notation.

eukaryote protein synthesis inhibitor and has been shown to increase EB production during chlamydial infections (41). Treatment with cycloheximide is thought to decrease competition between the host and *Chlamydia* for nutrients, allowing *Chlamydia* to replicate faster (42). To understand the impact of cycloheximide treatment on chla-

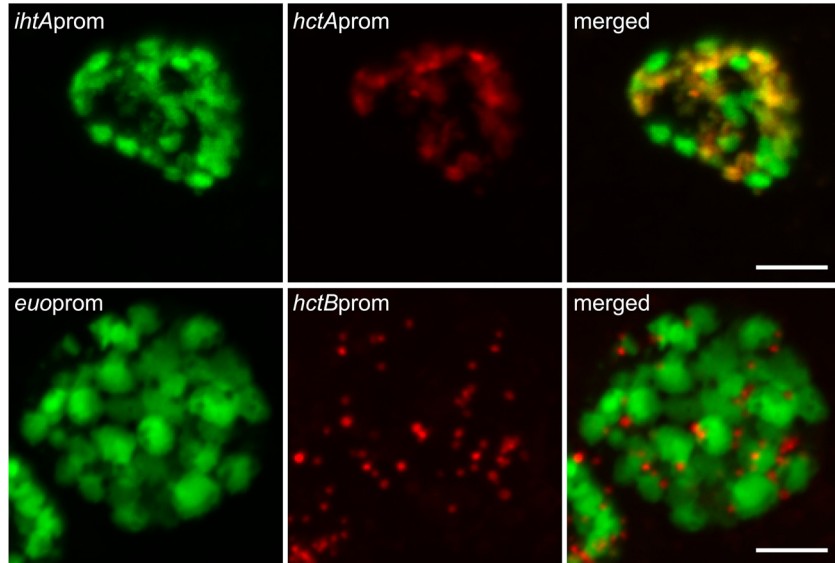

**FIG 9** Confocal fluorescence microscopy of cell type promoter expression upon inhibition of chlamydial division. Host cells were infected with *Ctr-hctA*prom-mKate2/*ihtA*prom-mNeonGreen (red and green, respectively) or *Ctr-hctB*prom-mKate2/*euo*prom-Clover (red and green, respectively), followed by treatment with penicillin (Pen) at 20 hpi. Samples were fixed at 24 hpi. Fixed samples were imaged by confocal microscopy, and maximum intensity projections are shown. Scale bars, 5 $\mu$m.

mydial developmental kinetics, the rates of RB and EB gene expression with and without cycloheximide were measured in individual inclusions for the entire cycle. Without cycloheximide treatment, the overall developmental pattern was retained; however, there was a delay in *euo*prom expression and a delay in the time to *euo*prom expression plateau (Fig. 10B, graphs 1 and 3). Additionally, EB gene expression in individual inclusions began later, and linear production had a significantly reduced slope (327 fluorescence units/h) in monolayers not treated with cycloheximide than in treated ones (482 fluorescence units/h) (Fig. 10B, graph 3). Interestingly, although *hctB*prom expression in the untreated cells increased at a linear rate until cell lysis, peak expression levels rarely reached that of the cycloheximide-treated cells, as cell lysis occurred before levels reached that of the treated inclusions.

These data further support that EB production is a property of the growth rate and is not likely a response to changing environmental signals. These data also suggest that growth rate of *Chlamydia* per cell is limited by steady-state levels of a limiting nutrient provided by the host, again indicating that EB development is unlikely to be linked to increasing competition or communication between *Chlamydia* but rather follows an intrinsic developmental program.

## DISCUSSION

The infection of vertebrate hosts by *Chlamydia* is dependent on the transition between two specific cell types, the RB and EB, that each have specialized functions. The RB undergoes cell division but is not infectious, while the EB form is responsible for mediating invasion of eukaryotic host cells and does not undergo cell division. The EB does, however, metabolize nutrients to maintain its infectious phenotype (18). This division of labor presents a critical dilemma for *Chlamydia*, as increasing cell numbers through RB division must be balanced with the production of infectious EBs. How *Chlamydia* regulates this balance is currently unknown.

Proposed mechanisms for the control of RB-to-EB development can be divided into two broad categories, a response to extrinsic environmental cues and an intrinsic developmental program. By developing mathematical models and running simulations of infection conditions, we determined that these two possibilities could be differentiated by generating competition between RBs for environmental signals or nutrients.

**FIG 10** Effect of cycloheximide on growth rate and EB production. Cos-7 cells were either treated with cycloheximide or vehicle only upon infection with *Ctr-hctB*prom-mKate2/*euo*prom-Clover. (A) Individual inclusion traces and averages of *euo*prom-Clover (RB) and *hctB*prom-mKate2 (EB) expression intensities monitored via automated live-cell fluorescence microscopy for cycloheximide (CHX)-treated infections. (B) Individual inclusion traces and averages of *euo*prom-Clover (RB) and *hctB*prom-mKate2 (EB) expression intensities monitored via automated live-cell fluorescence microscopy for vehicle (UNT)-treated infections. Purple lines are linear regression fits. Asterisks denote *P* value of <0.05. The cloud represents SEM. *y* axes are denoted in scientific notation.

To explore these models experimentally, we developed a live-cell reporter system to monitor cell type switching in real time at the single-inclusion level. Cell type-specific promoters were used to drive the expression of fluorescent proteins to monitor RB growth (*ihtA*prom and *euo*prom) and EB development (*hctA*prom, *hctB*prom, *scc2*prom,

mSystems®

and *tarp*prom). These promoter reporters were designed to detect spatial/temporal generation of fluorescence and the net of transcriptional, translational gene regulation, and maturation of the fluorophore and to not differentiate between these mechanisms. Chlamydial developmental kinetics observed using the live-cell reporter constructs were comparable to developmental data generated using qPCR for genome copies and reinfection assays to measure infectious progeny.

The use of live-cell promoter-reporters to interrogate cell type switching dramatically improved the resolution for monitoring chlamydial developmental transitions. Reporter expression was measured every 30 min at the single-inclusion level, which led to the identification of two different classes of late promoters. *hctB*, *scc2*, and *tarp* were all expressed ~22 hpi and, therefore, are considered a class of true late genes. However, our data suggest that *hctA* should be considered an early-late gene, as *hctA*prom-Clover expression is induced hours before the other late genes tested and responds differently to the inhibition of chlamydial cell division. This differential timing in expression between HctA and the late proteins is corroborated by our published RNA-seq data that demonstrated that the transcript encoding HctA was upregulated at 18 hpi, while the transcripts for HctB, Scc2, and Tarp were not detected until 24 hpi (18). Live-cell single-inclusion analysis also highlighted the inherent limitations of endpoint population-based assays. Single-inclusion dynamics demonstrated that kinetics of chlamydial development in single inclusions can be masked by cell lysis, superinfection, and reinfection in population-based studies.

Our live-cell data showed that competition for nutrients by increasing MOI and time delayed superinfections of both fusogenic and nonfusogenic inclusions, which generated competition for host cell and intrainclusion signals and did not alter time to EB development. These data strongly suggest that development from RB to EB is independent of a competitive intrainclusion or host environment but rather is responsive to one or more intrinsic cell-autonomous signals. Our data also showed that the developmental program is linked to a steady-state growth rate. *Chlamydia* grown at 35°C replicated slower and EB development was delayed compared to that of samples grown at 37°C. Conversely, *Chlamydia* incubated at 40°C replicated faster and EB development was initiated earlier than for growth at 37°C. Additionally, *Chlamydia* in cells treated with cycloheximide grew faster and EB development was initiated earlier than that for untreated cells.

Cell lysis and reinfection at late time points skewed the aggregate data, adding significant variability. The analysis of well-isolated single inclusions showed that each inclusion followed the same basic developmental profile. However, the *Chlamydia* in each inclusion had a unique growth rate. These data suggest that growth rate is set by steady-state kinetics in individual host cells, as EB gene (EB production) expression is linear in each cell until cell lysis but the slope varies between cells. This was also evident when comparing EB gene expression in cycloheximide-treated versus untreated host cells. The slope of *hctB*prom expression (EB production) is steeper with cycloheximide treatment, again suggesting that chlamydial growth rate is dependent on nutrient availability in the host cell. The linear kinetics of EB production suggests that *Chlamydia* does not encounter increasing nutrient limitation even toward the end of the cycle. The kinetics of chlamydial development within individual inclusions appears to mimic that of bacteria grown in a chemostat where replication rate is controlled by a limiting nutrient. Up to a point, the host cell is actively maintaining steady-state levels of nutrients that control chlamydial growth rate and that, in turn, control EB production rate.

In addition to growth, chlamydial cell division was also required to trigger EB development. Penicillin and DCS both target peptidoglycan synthesis at different points in the pathway, resulting in a block in cell septation during chlamydial replication (43). Both treatments, when added early in infection (prior to 14 hpi), inhibited EB formation, as measured by the production of infectious particles and expression of late gene promoter-reporters (*hctA*, *hctB*, *scc2*, and *tarp*). However, the effect of these drugs

on *hctA*prom-Clover expression differed significantly from the effects seen on *hctB*, *scc2*, and *tarp*. Although *hctA*prom-Clover expression was initially inhibited, expression was eventually initiated in the aberrant forms after an approximately 9-h delay. We speculate this delay is the result of gene dysregulation that, over time, produces spurious regulatory outputs. Pen addition at all times tested (2-h intervals from 16 to 28 h) resulted in an immediate overall increase in *euo*prom-Clover expression and an immediate overall decrease in *hctA*prom-Clover expression in inclusions compared to untreated samples. In contrast, *hctB*prom expression kinetics was similar to that of untreated controls for approximately 10 h after Pen addition, after which point further expression was inhibited. Additionally, *hctB*prom fluorescence was only evident in small cell forms, indicating expression was restricted to EBs, while *hctA*prom expression was evident in RB-like aberrant forms, suggesting expression in an intermediate cell form. These data suggest that inhibiting cell division blocks RBs from switching off *euo*prom expression and switching on *hctA*prom gene expression. However, if a cell is already committed to EB formation (*hctA*prom positive), EB gene expression continues (Pen insensitive) until the EB is fully mature (maximal *hctB*prom signal), which our data indicate takes about 10 h in *C. trachomatis* L2.

The treatment of *Chlamydia*-infected cells with penicillin, other antibiotics, or reagents that cause nutrient limitation results in a growth phenotype termed persistence (38). Persistence is characterized by aberrant RB forms that are larger than untreated RBs, do not undergo cell division, and do not produce infectious progeny (38). Although all these treatments cause aberrant RBs, the phenotypes vary (39, 44). Pen and DCS treatment cause persistence by inhibiting cell division through inhibiting peptidoglycan synthesis, while IFN-γ treatment causes persistence by inducing the enzyme indoleamine-2,3-dioxygenase in the host cell, which serves to deplete tryptophan levels in the cell, starving *Chlamydia* of this essential amino acid (39). Comparing the live-cell imaging data from these different persistence inducers revealed that the IFN-γ-treated *Chlamydia* never expressed Clover from any promoters tested early or late. This was also true for *Chlamydia* grown in the presence of the iron chelator bipyridyl. The *Chlamydia* from bipyridyl-treated infections never expressed the fluorescent reporters from early or late promoters. This dramatic difference in gene regulation suggests different mechanisms are involved and that persistence is not a phenotype associated with a specific gene expression profile.

Overall, our data support a model in which RB-to-EB development follows a cell-autonomous preprogrammed cycle that requires chlamydial division. Our initial mathematical models assumed an inhibitory signal that, at high concentrations, inhibited RBs from differentiating into EBs. The concentration of this signal was depleted by metabolic utilization, and RB-to-EB differentiation occurred. We have now updated this model to reflect our current data supporting an intrinsic signal linked to chlamydial growth rate and cell division. This model suggests the involvement of an internal signal in the nascent RB that, at high concentrations, inhibits RBs from differentiating into EBs, and that the signal concentration is depleted through dilution via 3 to 5 cell divisions and not metabolic utilization. After the inhibitory signal is reduced below a threshold, RBs are capable of transitioning to EBs (Fig. 11). Of the current proposed models in the literature (nutrient limitation [45], inclusion membrane limitation [46], and RB size [14]), only the model based on RB size is consistent with our data. The RB size model described by Lee et al. proposed that RB growth rate is lower than the division rate, leading to a size reduction (depletion of signal) of the RBs after each division. After several rounds of division, a size threshold is reached and EB development is triggered (14). This proposed mechanism fits our model, as size would act as the inhibitory signal that is reduced through cell division. It should be noted that although we propose the dilution of an inhibitor as the intrinsic signal to control cell type switching, it is equally possible that a positive signal linked to cell division, such as the development of asymmetry/polarity, could act as an EB-promoting signal.

Chlamydial development can be considered to occur in two steps, an RB exponential growth step starting ~12 hpi (*C. trachomatis* serovar L2) and an asynchronous EB

## Gene expression profile

[ Signal ] (3-5 Divisions)

RB (*euo/ihtA*prom)

IB (*hctA*prom)

EB (*hctB/scc2/tarp*prom)

**FIG 11** Schematic of concentration-dependent RB/EB conversion model. The schematic shows diminishing signal concentration within RBs (dark to light blue) upon cell division. Depletion of the signal permits RBs to produce IBs (red), which then convert to EBs (orange). RB$_R$s divide into two subsequent RBs. RB$_E$s are competent to make EBs and divide into a RB and an IB. Each cell form has predicted associated promoter expression phenotypes. RB (RB$_R$ and RB$_E$), *euo-ihtA*; IB, *hctA*; EB, *hctB-scc2-tarp*.

production step starting at ~18 hpi (*C. trachomatis* serovar L2) (47, 48). Although the size reduction model and our model explain some of the gene expression patterns that control cell type switching, it is clear that EB development is more complicated than these simple switch models. The output of the models fit the switch between the RB exponential growth phase and the beginning of EB development, but they do not adequately explain the continued requirement for cell division during asynchronous EB production. Our data show that Pen treatment blocks the *euo*-to-*hctA* gene expression switch even when added late in infection (28 hpi), well after the time of initial EB formation (~18 hpi). Further evidence for a dilution-independent second step is the observation that the *euo*prom-to-*hctA*prom switch is initially blocked by both DCS treatment and Pen treatment, yet this inhibition is eventually overcome and *hctA*prom-Clover is expressed after a 9-h delay. Unlike Pen treatment, where RBs continue to increase in size, DCS impacts cell growth, resulting in smaller RBs and, thus, limiting the effect of dilution. These observations support a second developmental regulatory step that is independent of inhibitor dilution, suggesting cell division itself is an important step in committing to the EB cell type.

Our interpretation of these data is that EB formation is multifactorial and requires multiple steps to form a final infectious EB. The first step is the loss of the inhibitory signal in the RB through multiple rounds of division, where early RBs (RB$_R$) divide 3 to 5 times by binary fission, eventually becoming competent to produce EBs (RB$_E$). This is followed by a second step that is dependent on asymmetric cell division creating two cells with different expression profiles. One daughter cell remains an RB$_E$ (*euo*prom positive), and the second daughter cell becomes committed to EB formation (IB, *hctA*prom positive) (Fig. 11). The committed IB cell (*hctA*prom positive) does not divide but matures into the infectious EB (*hctB*prom, *scc2*prom, and *tarp*prom positive).

Further divisions of the $RB_E$ cell produce one $RB_E$ and one IB leading to the linear increase in EBs that we report. The data from the Pen treatment experiments also suggest that EB maturation, from *hctA*prom positive to *hctB*prom positive, takes ~8 to 10 h, but we do not yet know when, along this progression, infectivity is gained.

Additional support for asymmetric EB production is the observation that *hctB*prom signal (EB production) follows a nearly perfect linear trajectory and is not logarithmic during the EB production phase (24 hpi; cell lysis) (Fig. 10A, Fig. S5, and Movie S2). In contrast, the *euo*prom signal (RB growth) transitions from log to linear to no growth (Fig. 10A and Movie S2). These observations suggest that the $RB_R$ cell population expands by exponential growth followed by a transition to the $RB_E$ cell type. The $RB_E$ then divides asymmetrically, leading to EB production with no gain in $RB_E$ numbers. Asymmetric cell division producing two cells with differing fates is reminiscent of stalk/swarmer cell systems best described in *Caulobacter crescentus* (49) but also described in the *Planctomycetes* genus that is more closely related to *Chlamydia* (50). This is also supported by other studies that have provided evidence for asymmetric cell division in *C. trachomatis*. These studies show that the cell division machinery assembles asymmetrically, leading to polarized RB division (43, 51, 52). Additionally, the EB itself is asymmetric, demonstrating hemispherical projections that can be seen by electron microscopy (53).

Overall, our data show that the combination of mathematical modeling and live-cell gene reporter imaging is a powerful tool to tease apart the molecular details of cell type development. Continued revision and testing of our models of development will lead to an expanded understanding of cell type development in this important human pathogen.

## MATERIALS AND METHODS

**Organisms and cell culture.** Cos-7 and HeLa cells were obtained from the American Type Culture Collection (ATCC). Cos-7 cells were used for all experiments unless otherwise specified. Both Cos-7 and HeLa cells were maintained in a 5% $CO_2$ incubator at 37°C (unless otherwise indicated) in RPMI 1640 (Cellgro) supplemented with 10% fetal plex and 10 g/ml gentamicin. All *C. trachomatis* L2 (LGV 434) strains were grown in and harvested from Cos-7 cells. Elementary bodies were purified by density centrifugation using 30% MD-76R 48 h postinfection (18). Purified elementary bodies were stored at −80°C in sucrose-phosphate-glutamate buffer (10 mM sodium phosphate [8 mM $K_2HPO_4$, 2 mM $KH_2PO_4$], 220 mM sucrose, 0.50 mM L-glutamic acid; pH 7.4). *Escherichia coli* ER2925 (mutated in *dam* and *dcm*) was utilized to produce unmethylated constructs for transformation into *Chlamydia*.

**Reporter plasmids.** The backbone for all promoter-reporter constructs was p2TK2SW2 (54). Promoters were amplified from *C. trachomatis* L2 genomic DNA using the primers indicated (see Table S1 in the supplemental material). Each promoter sequence consisted of ~100 bp upstream of the predicted transcription start site for the specified chlamydial genes plus the untranslated region and the first 30 nt (10 amino acids) of the respective ORF. Promoter sequences were inserted into p2TK2SW2 downstream of the ColE1 ORI. Fluorescent reporters (EGFP/Clover/mNeonGreen/mKate2) were ordered as gene blocks from Integrated DNA Technologies (IDT) and inserted in frame with the first 30 nt of the chlamydial gene. Each ORF was followed by the *incD* terminator. The *bla* gene was replaced by the *aadA* gene (spectinomycin resistance) from pBam4. The final constructs reported in this study were p2TK2-*ihtA*prom-EGFP, p2TK2-*hctA*prom-Clover, p2TK2-*hctB*prom-Clover, p2TK2-*scc2*prom-Clover, p2TK2-*euo*prom-Clover, p2TK2-*tarp*prom-Clover, p2TK2-*hctB*prom-mKate2/*euo*prom-Clover, and p2TK2-*hctA*prom-mKate2/*ihtA*prom-mNeonGreen.

**Chlamydial transformation and isolation.** Transformation of *C. trachomatis* L2 was performed as previously described (54) and selected using 500 ng/µl spectinomycin. Clonal isolation was achieved via successive rounds of inclusion isolation (MOI, <1) using a micromanipulator. The plasmid constructs were purified from chlamydial transformants, transformed into *E. coli*, and sequenced.

**Infections.** To synchronize infections, host cells were incubated with *C. trachomatis* EBs in Hanks' balanced salt solution (HBSS) (Gibco) for 15 min at 37°C with rocking. The inoculum was removed and cells were washed with prewarmed (37°C) 1 mg/ml heparin sodium in HBSS. The HBSS with heparin was replaced with fresh RPMI 1640 containing 10% fetal bovine serum, 10 µg/ml gentamicin, and 1 µg/ml cycloheximide, unless otherwise stated. For cell division experiments, chlamydial cell division was inhibited by the addition of 1 U/ml penicillin G or 40 µg/ml D-cycloserine to the media. To starve *Chlamydia* of tryptophan, HeLa cells were incubated for 24 h in medium containing 2 ng/ml recombinant human IFN-γ (PHC4033; Invitrogen) prior to infection. Iron starvation of *Chlamydia* was achieved by treating Cos-7 cells with the iron chelator bipyridyl (100 µM) upon infection with Ctr-L2-prom EBs (55).

**Replating assays.** Ctr-*hctA*prom-Clover EBs were obtained from infected Cos-7 cells by scraping the host monolayer and pelleting via centrifugation for 30 min at 17,200 relative centrifugal force. The EB pellets were resuspended in RPMI via sonication. For reinfection, Cos-7 cells were plated to confluence in clear polystyrene 96-well microplates. EB reinfections consisted of 2-fold dilutions. Spectinomycin was

added to superinfection experiments to prevent wild-type *C. trachomatis* L2 growth. Infected plates were incubated for 29 h. Cells were fixed with methanol and stained with 4′,6-diamidino-2-phenylindole (DAPI). The DAPI stain was used for automated microscope focus and visualization of host cell nuclei, and GFP-Clover was used for visualization of EBs and inclusion counts. Inclusions were imaged using a Nikon Eclipse TE300 inverted microscope utilizing a scopeLED lamp at 470 nm and 390 nm and BrightLine bandpass emissions filters at 514/30 nm and 434/17 nm. Image acquisition was performed using an Andor Zyla sCMOS in conjunction with μManager software. Images were analyzed using ImageJ software (56) and custom scripts.

**Genome number quantification.** Chlamydial genomic DNA was isolated from infected host cells during active infections using an Invitrogen PureLink genomic DNA minikit. An ABI-7900HT reverse transcription PCR system was utilized for the quantification of genomic copies. A DyNAmo Flash SYBR green qPCR kit and *hctA*-specific primer were used for detection.

**Fluorescence microscopy.** Cos-7 monolayers were infected with synchronized *Ctr*-L2-prom EBs. Live infections were grown in an OKOtouch $CO_2$/heated stage incubator. Infections were imaged using a Nikon Eclipse TE300 inverted microscope using epifluorescence imaging and a 20×, 0.4-numeric-aperture objective, giving a depth of field of about 5.8 μm. A ScopeLED lamp at 470 nm and 595 nm and BrightLine bandpass filters at 514/30 nm and 590/20 nm were used for excitation and emission. DIC was used for focus. Image acquisition was performed using an Andor Zyla sCMOS camera in conjunction with μManager software (57). Images were taken at 30-min intervals from 10 to 48 h after *Ctr*-L2-prom infection unless otherwise stated. Live-cell infections were performed in 24- or 96-well glass-bottom plates, allowing treatments to vary between wells. Multiple fields were imaged for each treatment. Fluorescent intensities for individual inclusions were monitored over time using the Trackmate plug-in in ImageJ (22). Inclusion fluorescent intensities were then analyzed and graphed using pandas, mat-plotlib, and seaborn in custom Python notebooks. The scripts for this analysis are available from the github account (https://github.com/SGrasshopper).

For confocal microscopy, samples were fixed with 4% paraformaldehyde, washed with phosphate-buffered saline, and mounted with MOWIOL. Confocal images were acquired using a Nikon spinning disk confocal system with a 60× oil immersion objective, equipped with an Andor Ixon electron-multiplying charge-coupled device camera under the control of Nikon Elements software. Images were processed using the image analysis software ImageJ (http://rsb.info.nih.gov/ij/). Representative confocal micrographs displayed in the figures are maximal intensity projections of the three-dimensional data sets unless otherwise noted.

**Data availability.** All data, bacterial strains, and methodologies are available upon request.

## SUPPLEMENTAL MATERIAL

Supplemental material is available online only.

**MOVIE S1**, MOV file, 4.7 MB.
**MOVIE S2**, MOV file, 0.5 MB.
**FIG S1**, PDF file, 0.1 MB.
**FIG S2**, TIF file, 0.3 MB.
**FIG S3**, TIF file, 1.1 MB.
**FIG S4**, TIF file, 0.7 MB.
**FIG S5**, TIF file, 2.2 MB.
**TABLE S1**, TIF file, 2.7 MB.

## ACKNOWLEDGMENTS

We thank Dan Rockey at Oregon State University for supplying the isogenic *C. trachomatis* serovar J and Js strains.

This work was supported by NIH grants R01AI130072, R21AI135691, and R21AI113617. Additional support was provided by start-up funds from the University of Idaho and the Center for Modeling Complex Interactions through their NIH grant, P20GM104420.

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
