## [Reviewer comments · mSystems]

Single Inclusion Kinetics of *Chlamydia trachomatis* Development

Travis Chiarelli, Nicole Grieshaber, Anders Omsland, Christopher Remien, and Scott Grieshaber

Corresponding Author(s): Scott Grieshaber, University of Idaho

Review Timeline:

Submission Date:	July 21, 2020
Editorial Decision:	August 24, 2020
Revision Received:	September 1, 2020
Editorial Decision:	September 11, 2020
Revision Received:	September 14, 2020
Accepted:	September 24, 2020

Editor: Michael Rust

Reviewer(s): The reviewers have opted to remain anonymous.

Transaction Report:

DOI: <https://doi.org/10.1128/mSystems.00689-20>

Reviewer comments:

We thank the reviewers for their constructive criticism and appreciate their efforts. Specific responses to major concerns are described below.

Reviewer #1 (Comments for the Author):

The article by Chiarelli and colleagues uses promoter-specific single bacterial cell fluorescence imaging to examine the nature of signals associated with chlamydial intracellular growth. The article addresses a number of larger questions including the nature of the molecular signal that leads to developmental transitions, the connection between antibiotic treatment and development, and the gene-specific transcriptional kinetics in differently treated, infected host cells. The article is written and assembled very well.

The manuscript begins with the development of a very nice system for quantifiable analysis of developmental gene expression at the level of single chlamydiae. The gene choice is excellent and the analysis is very interesting. The system will be useful for a variety of subsequent studies directed both at gene-specific expression and larger cellular processes. I expect that interest will be high among researchers in the field.

Specific comments:

1: It would be useful to provide a descriptive figure showing the connection between the tools of data collection (i.e. fluorescent micrographs that are then quantified) and the generation of graphs. This would be helpful to readers as they work to assimilate and correlate the considerable amount of data in the document. Perhaps one way to do this would be to include a microscopic field of view containing a limited number of live infected cells, with images of this field collected through the developmental cycle. A labeling strategy as used in figure 11 may be a good model. A graph of data representative of the label development would be included in this figure. It could be a data figure and not just for experimental explanation, but it could be set up to instruct readers as to how the microscopy directly connects to the data graphs.

We have added a movie figure showing fluorescent micrographs over time as well as the fluorescent intensity traces. We have also added additional detail to our materials and methods and added a reference to our recently published techniques paper which has more detail on our system.

2: It is also possible that the overall message would be enhanced by making the focus a bit more general and not limiting the primary objective to the "intrinsic vs extrinsic" question. First of all, the manuscript does not focus on this single issue: the authors do a great job of looking at different aspects of chlamydial development using their novel model system. The intrinsic/extrinsic question is one part of the overall message, which includes interesting gene regulatory questions (hctA vs hctB for example) and the effects of persistence inducers on gene expression.

We changed the title to reflect the direction the data indicated "Single inclusion kinetics of chlamydial development"

3: The logic behind the mathematical model is interesting and well described. It might be useful to briefly discuss how other systems integrate different signals into similar, major developmental transitions. For example, Bacillus sporulation involves a multi-factorial sensing cascade, because the effect of the decision is so extreme. In the chlamydial system, is it not possible that both intrinsic and extrinsic signals are required to facilitate commitment to the final steps in developmental differentiation in chlamydiae?

We added additional comparisons to *Caulobacter crescentus* development in the discussion as our model matches that system at the latter stages of development

Overall, the data are excellent, the depth of study is impressive, and the figures are well assembled.
Thanks

Some minor points that should be addressed:

4: Legend for Figure 1: "I", "S", and "E" are defined, "R" should also be defined. Alternatively, just use RB, IB, and, somehow, EB in the images themselves.

Fixed

5: In Figure 1: the box included in each panel that defines the individual lines should only be included once in the figure, or the colors can simply be defined in the legend.

Fixed

6: Several places in the figures: the small boxes that state "1e4... 1e5" etc., are not defined in a clear way- it is likely a titer of bacteria, but that might not be correct. But, and importantly, it needs to be explained why they are either the same or different in these experiments.

Fixed

The following comments address items that can be addressed at the discretion of the authors.

*The abbreviation for *C. trachomatis* should be spelled out, not abbreviated Ctr. It would just make reading easier, as there are quite a few three letter acronyms already.*

Fixed

Lines 189-191: Combine these sentences into a single sentence saying that none of these promoters varied as a function of MOI.

Fixed

Line 247: ... compared to untreated cells.

Fixed

Line 251 and elsewhere: What do the authors think is going on with the lag in expression in these conditions? Does it fit the model in any way?

We really have no idea why HctA turns on again after the ~9h delay. We speculate some sort of gene dysregulation, perhaps through degradation of regulators. It does not fit into our model currently.

Line 262: It is not clear what the GFP florescence is telling the reader, or what the source of the GFP is.

Fixed

Line 273: It might be better to state that pen induces aberrancy that leads to persistence, and not the other way around.

Fixed

Lines 282-3: It is quite challenging to assess, and perhaps very interesting, that these genes do not behave similarly in cells treated with ifnG vs. beta lactams. I don't know if a more direct comparison could be made. The text on lines 373 thru 377 addresses these differences, but it would be great if this was stressed somehow in the results.

We added bipyrindyl as a treatment and saw a similar phenotype suggesting a block in protein expression leading to no cell growth or cell division while beta lactams block cell division but not cell growth.

Line 294: the sentence beginning with "Pen treatment blocks..." can be deleted as it is stated in prior text. Perhaps the phrase related to "large cells" could be included on line 273.

Fixed

Lines 306-307: the concept about how the bacteria divide could be deleted. It is interesting but not relevant to the message.

Fixed

Legend for figure 2: the sentence beginning on line 551 should be deleted and the concept included in the results.

Fixed

Figure 6 and 7 could be combined and redundant panels removed. The ifu and genome copy panels should be next to each other. Within a figure, a single color should mean the same thing whenever possible.

Fixed

Figure 8: the collected fluorescence images are very nice and perhaps should be placed in a figure at the beginning of the document.

Fixed with our movie of dynamics

Figure 9: Could these data be included in a big figure with the beta lactam/cycloserine data?

Combined IFN-G and Bpdl in the same figure.

Figure 10 is very interesting stuff. Well done!

Thanks

Reviewer #2 (Comments for the Author):

Comments for Authors:

The authors sought to test two models of RB-to-EB transitioning: an "intrinsic" cell autonomous program inherent to each RB or an "extrinsic" environmental signal to which RBs respond. Utilizing a fluorescent reporter approach coupled to developmental-specific promoters and live cell imaging, the authors state that EB production was not influenced by the initial starting inoculum (increased MOI / superinfection), suggesting an intrinsic program not influenced by environmental factors. Additionally, they claim that their reporter assay indicates that EB development is a multistep process linked to RB growth rate and cell division, a widely-held view in the field and supported by previous studies. They also found *euo* and *ihfA* promoters were active in RBs, *hctA* in 'early EBs/intermediate cells' and *hctB*, *scc2*, and *tarp* active in the 'maturing' EBs.

While their results are interesting and represent a new way of studying Chlamydia's developmental cycle, care needs to be taken when interpreting the results presented in this study. The deficiencies in the proposed models / techniques need to be more thoroughly addressed, important controls need to be added, and generalizations based on their observations need to be more carefully scrutinized. The assumptions built into each model and into the interpretation of the reporter assay need to be more clearly stated and backed up with reasoning based on the primary literature, particularly regarding the chlamydial developmental cycle. This is the case both for the proposed 2-model testing approach as well as the interpretation of data generated by the promoter reporter assay.

Major comments:

Environmental vs. intrinsic modeling:

- Why do the authors assume that an EB-inhibitory signal exists, rather than the absence of an RB-promoting signal? They mention other potential mechanisms in their discussion, but they need to make a stronger case for why these two models were chosen for investigation.

Although this was mentioned in the discussion we have made it more clear in the results. The signal could be a promoting or inhibitory signal, the sign would just change in the model. We did not test which of these is most likely. Although any complex regulatory network contains elements of both positive and negative regulation.

- Neither model appears to adequately address the asynchronous nature of the RB-to-EB conversion, a well-documented phenotype known to occur during the developmental cycle.

Our final proposed model does actually address the asynchronous nature of the developmental cycle. Asynchronous development refers to the observation that after initial EB development throughout the rest of the cycle there are both RBs and EBs present in the same inclusion. Our asymmetrical cell division model has RBs (RB_E) present throughout the cycle producing EBs.

- An additional issue with the modeling is that neither appears to fit well with what is known about Chlamydia development. Both models assume that the eventual number of genomes will be the same regardless of the model chosen or replication rates employed. This is not supported by the primary literature. Decades of research has shown that numerous factors influence EB production in host cells, the most obvious being competition with the host cell for nutrients, which is significantly reduced with cycloheximide treatment, resulting in higher numbers of EBs generated per cell over time.

We have added an additional figure showing cycloheximide and temperature effects on single inclusions late in infection. Our single inclusion analysis shows that the life cycle of each inclusion follows the same developmental profile kinetics. However each inclusion appears to have a unique growth rate and the time to cell lysis is also variable leading to the average of the inclusions to demonstrate kinetics different from each individual. This data strengthens our model that the cycle is pre-programmed and cell autonomous but growth rate dependent. This growth rate appears to be set by steady state kinetics in each cell as EB production is linear until cell lysis. The slope of EB production responds to cyclo treatment suggesting that nutrient availability to Chlamydia is dependent on nutrient availability in the cell. However, nutrient availability appears to be a steady state property of the cell set by the cell's growth conditions and Chlamydia does not appear to encounter nutrient limitation at the end of the cycle.

- The models would appear to do a better job of comparing the very first instances of the initiation of conversion, rather than actual rates of conversion. Neither model addresses the possibility that, just as the RB division process can proceed at different rates during development, so too might the RB to EB conversion process. Because conversion is asynchronous, the rate of conversion within a population is a far better indicator than when the first indicators of conversion are observed.

We agree. We initially focused on the initiation of EB production. However we have now added additional data looking at the end of the cycle in individual inclusions.

Promoter-reporter system:

- it is unclear whether all infections in these studies were carried out via centrifugation (to synchronize infections) and whether cycloheximide is used for all infections throughout the manuscript. These two factors will directly influence the interpretation of the results. They need to be made clear in the methods section, in the text, and in Figure legends.

We synchronize using a heparin wash. We have added more detail and references to our synchronization method. We have also added a figure comparing development with and without cycloheximide.

- The fluorescent reporter assay appears to be dependent on a steady state level of protein synthesis. As such, some type of control is necessary to eliminate the possibility that apparent changes in promoter activity are not simply the result of general changes in translation brought about by the experimental conditions being tested. Assuming different developmental forms of Chlamydia also have different rates of protein synthesis, this could also severely impact data interpretation.

As we analysed individual inclusions under a variety of conditions that affect growth rate (temperature, super infection, MOI, cycloheximide), we do not agree that variations in protein synthesis would impact our interpretation of the data. The reporters all acted as would be expected across these conditions.

- The data generated by the authors would be more interpretable if inclusion size is removed as a potential compounding variable. The authors report their data readout as 'inclusion fluorescent intensities'. It should be

stated whether this refers to 'mean' fluorescent intensity values (average pixel intensity per inclusion) or 'integrated' fluorescent intensity values (the summation of all pixel intensity scores per inclusion). Having both values would allow the authors to assess whether inclusion size drastically affects their readouts (and subsequent result). As presented, the current study does not appear to present a control for inclusion size when accessing their measurements. This is critically important if no attempt was made to synchronize infections by utilizing centrifugation at the start of each assay, however, even when centrifuged, Chlamydia and their inclusions often mature at different rates with cells.

We measure the fluorescence intensity from each reporter in each inclusion. Inclusion size is not a factor as we are looking at the kinetics of each inclusion. In our system the ROI does not change size (it is made to encompass the entire inclusion at the end of imaging) so mean intensity and total (mean * area) would always be directly proportional and would generate the same kinetic curves. Again, the infections were synchronized, we follow the kinetics of individuals as well as present the average of all individuals. Yes we do see variation in growth rate of individual inclusions but this variation is to be expected. This has been highlighted in a new movie figure.

- How does the fact that fluorescence intensity is measured at the level of inclusions and not individual RBs affect the interpretation of the results? The assay, as described, appears incapable of differentiating between inclusions containing a few bright RBs and those containing a number of moderately bright RBs, likely the reason why maximal intensity measurements do not appear to differ between MOI infection groups. Additionally, effects of general protein translation upon super-infection / enhanced MOIs could also explain differences, and thus constitutive promoters should be used as controls to eliminate this possibility.

Measuring gene expression from individual chlamydial cells is not currently possible due to their small size and dynamic movement within the inclusion.

- Were images taken as Zstacks or single imaging planes chosen? If images represent single imaging planes, how did the authors decide which fields of view to image for all of their fluorescence imaging studies? If an autofocus program was used, what were its parameters? Field of view is important as it will directly affect the apparent 'size' and 'brightness' of an inclusion that is imaged and thus its mean fluorescence intensity. A control experiment for each construct showing the degree to which field of view influences overall mean fluorescence reported for individual inclusions would be an appropriate control if Zstacks were not generated.

We used epifluorescence imaging and a 20X 0.4NA objective giving a depth of field about 5.8 μm , collecting most of the light from each inclusion. As inclusions increased to very large sizes we did see some variation in EUOprom fluorescent intensity as some RBs moved in and out of the focus plane. We have added a reference for our methods in the methods section.

- How do the authors explain the variance in relative fluorescence intensities at later time points for each fusion construct? Some of the ranges at later time points appear to be substantial (~1 log).

Our initial studies were focused on the initiation of RB to EB conversion. We have now added results analysing the late time points of infections and focused on birth to death of inclusions. The variability late in infection is a result in the variability in the lysis of the host cell, this then reduces the number of individual inclusions included in the mean and confidence interval curves increasing variability. Our new section on late inclusion dynamics addressed this issue by focusing only on inclusions that we could verify to capture the entire growth cycle, i.e. initiation of Euo expression until cell lysis.

- How did the authors determine the thresholds utilized to calculate 'relative fluorescence' for their assays? Ideally, some background signal should be reported for either uninfected cells or cells infected with a non-fluorescent strain so that readers are able to assign a corresponding level of confidence in the values reported, particularly at the lower end of the detection range.

We simply recorded fluorescence. We set no threshold. We did however use baseline subtraction as there is some variability in the excitation intensity across the field of view. But since we are focused on the kinetics of each inclusion we don't believe this had any effect on our results. Also readers can now judge the effects of illumination variation themselves in the supplemental movie showing a field of view alongside the intensity plots.

*Line 45-48: "The formation of EBs followed a cell type gene expression progression with the promoters for *euo* and *ihfA* active in RBs, while the promoter for *hctA* was active in early EBs/intermediate cells and finally the promoters for the true late genes, *hctB*, *scc2*, and *tarp* active in the maturing EB."*

- The authors need to state how they are validating their results and how they distinguish between RBs, 'early EBs/intermediate cells' and 'EBs'. For the RB / EB categories size differences appear to be sufficient, however, this does not appear to be the case for the middle category

- The authors need to better define what a 'maturing EB' is.

As there is no direct data correlating size with infectivity, size is also not the best measure of an "EB". Our data and model suggests that the Chlamydia infection proceeds EB (no gene expression)->RB (EUOprom)->IB (HctAprom)->EB(HctBprom). We do not currently know when infectivity is gained in this process. From both our confocal images (EUOprom, HctAprom, HctBprom) expression occurs in separate cells and from the kinetic data EUO->HctA->HctB and the pen and DCS data all suggest a progression. We state clearly that we are calling cells that express EUO RBs while cells expressing HctA are IBs and cells expressing HctB are EBs. We have added in the discussion that we do not know for certain that the HctA positive cells that are not yet expressing HctB are infectious as they do not yet express the EB typeIII effectors (Tarp or the chaperone Scc2) it is unlikely the HctA expressing cells are infectious.

Line 41-43, 56-57: "We demonstrate that RB to EB development follows a cell autonomous program that does not respond to environmental cues."

- This is an overgeneralization. The authors focus primarily on the initiation of transcriptional cues that eventually result in conversion. Their focus in most of their reported results (Fig 4-5) focuses primarily on the early stages of the developmental cycle and curiously their IFU data is often reported only to 24 hrs, a time point in which their inclusion forming unit assay appears to be close to its limit of detection and before actual nutrient depletion resulting from superinfection and high MOI infections is expected to occur.

We have added a section in the results looking at the entire life cycle of inclusions, birth to death. This new section further supports our model of a cell autonomous program that controls cell type progression. We also further show that this is likely growth rate dependent. As our data highlight, MOI experiments can only measure what is happening at the population level. Our data suggest that this population level data obfuscates important aspects of chlamydial development

Line 99-101: "We show here that neither the limiting membrane hypothesis nor environmental nutrient limiting mechanisms are consistent with our experimental results, and that EB development likely follows a cell autonomous program."

- Under the conditions used in this assay, nutrient limiting conditions would not arise until later time points in infection. This would be exacerbated if the authors used cycloheximide during these infections, as limiting host

cell protein synthesis would make nutrients even more plentiful within infected cells and push back any potential nutrient limitations until later in the developmental cycle.

We have addressed this with the birth to death inclusion analysis of cells with and without cycloheximide. Interestingly this led to the observation that the cell is acting like a chemostat and Chlamydia don't seem to encounter limiting nutrient conditions.

- The authors base the statement on the assumption that a host cell is incapable of meeting the early metabolic requirements of 32 independently-developing EBs until 24 hpi (Fig 4-5). In general, cells are capable of maintaining much larger numbers of Chlamydia RBs and EBs late in the developmental cycle (40-48 hpi), so these conditions do not appear to be appropriate for testing the effects of nutrient limitation on RB-EB development early in the developmental cycle.

No, we made no such assumptions; we simply asked the question if the initial initiation of RB to EB development was dependent on competition inside the cell as has been proposed by multiple previous studies. We have added further results looking at late stages of development and these results add further support to the cell autonomous model and suggest that chlamydia never encounters increasing nutrient limitation but instead lives in a consistent nutrient state.

- It is telling that EB development appears to begin to diverge in both superinfections and high MOI infections after 24 hours (Fig 4-5), roughly corresponding to the time point in development when chlamydial inclusions fuse, and when nutrient limitation in an infected cell would most likely become significant.

We investigate this with our late birth to death data analysis and show that this divergence is due to variation in host cell death and confounded by both super infections and reinfection of neighboring cells. Also, fusion as a factor in development was addressed in the C. trachomatis-J and Js experiments and showed very little difference between superinfections.

- In order to accurately investigate the effects of nutrient depletion on the chlamydial developmental cycle pre-24 hpi, nutrient-deplete conditions should be generated utilizing techniques previously described in the literature with regard to amino acid / iron starvation. For example, if nutrient limitation plays no role in the RB-EB conversion, then MOI / superinfection should have no effect on EB development under artificially-produced nutrient limitation using interferon or iron chelators (in the absence of cycloheximide) at concentrations just below inhibitory concentrations for infections carried out at an MOI of 1. Alternatively, differences between cells left untreated or treated with cycloheximide could be compared, as it would be predicted that untreated cells would achieve nutrient depletion earlier than treated cells.

We are not directly investigating the role of nutrient depletion on chlamydial development. Although our data support our hypothesis that growth rate directly affects development. Further studies involving nutrient depletion will be a future focus of our investigations.

Figure 1D:

- The labeling is confusing. Better to state '1/2 RB doubling time' and '2x RB doubling time'.

Changed

- Do the authors mean the doubling rate? A distinction needs to be made between RB 'growth rate' and 'replication rate'. Growth vs. division are not mutually exclusive in Chlamydia. The two are potentially

decouplable, given penicillin-treatment data and the size-dependent model of RB-to-EB differentiation proposed by Lee et. al. Rapid division in the absence of a corresponding enhancement in cell size would result in smaller cells, which has been previously reported to be a necessary phenomenon prior to conversion. Similarly, a decline in replication while growth continues unabated would result in larger RBs, thus effectively preventing or pushing back conversion.

We did not measure cell size. The simplest interpretation of our data is asymmetric cell division (EB factory). A more complex interpretation would be a system that balances a size threshold vs growth rate to always produce a continuous linear formation of EBs until cell lysis. This will be the focus of additional studies.

Figure 2:

- In the legend the authors state that infections were 'synchronized' but do not refer to the method of synchronization either in the text or in the methods.

Added in material and methods and added a reference.

- Data are presented as 'relative fluorescence'. What does that mean? The axis on each graph is different, likely due to the relative strength of each given promoter? Should the authors assign a higher confidence in the reporters whose baseline 'relative fluorescence' intensities are higher.

Fixed

- For Fig. 2B, why are IFU counts represented by a linear axis and not a logarithmic one?

Because our data suggest EBs are produced at a linear rate.

- Line 551-553: Is the initiation of fluorescence fairly uniform or does it only appear that way due to the limit of detection inherent to the assay? It is telling that the variation in the data almost always increases substantially at higher relative fluorescence intensities.

While we don't claim single molecule detection, the detection of high quantum yield fluorescent proteins is very efficient. We have presented additional data following inclusion birth to death and show that the majority of the variation is introduced by cell death, super-infections and reinfections.

Line 151-153: "We measured >50 individual inclusions per strain and found very little inclusion to inclusion variability in the timing of initiation of expression."

- While the initiation of promoter activity does appear to be relatively uniform, I would disagree with the authors regarding the later in development each inclusion is measured. As the authors have a rather large data set available, it shouldn't be very difficult to come up with statistics for each promoter expression construct per time point measured. This would provide a confidence level for each reporter throughout development.

Our new data focusing on late gene expression/EB production highlight that each cell appears to set a growth rate for Chlamydia within the inclusion.

Figure 3:

- Line 165-166: "The lower replication rate at 35°C was also reflected in the qPCR genome counts (Fig. 3B)." o This is not well-reflected at early time points. It would be nice to know at what time point genome copy # differs significantly between the 3 temperatures. From the graph, it appears they begin to differ around 12-13 hpi. Just before that, you see large differences in ihtAprom but almost no difference in genome copy number.

qPCR from populations is inherently noisy and not easily compared to gene expression in a single inclusion over time, which is why we developed this non destructive method to assess development in single inclusions. Any destructive population based method, especially late in infection, would not accurately capture the kinetics of individual inclusions and would not generate a useful comparison as some cell lyses, super-infection and reinfection occurs confounding analysis by population based methods.

- How do the authors know that differences in expression are not due to the result of the effects of temperature on protein translation in general? This would likely account not only for differences in the initiation of promoter activity but also for the apparent differences in the rate at which signal intensity increases. A control construct with a constitutive promoter showing no effect of temperature on the timing of fluorescence induction seems warranted.

The effect of temperature on bacterial growth is likely a combination of kinetic effects, including gene expression, leading to the overall general observation that bacterial growth follows a linear relationship between temperature and the square root of growth rate. This is of course excluding specific toxicity issues at the extreme ends of temperature.

*- Given the rapid spike in *hctA*prom activity at the 40C temperature (Fig 3C), I would have expected a correspondingly large divergence in IFU counts, starting at ~15 hpi (Fig 3D). Please provide the full data for the IFU counts over the entire 40-48-hour developmental cycle; given the present axis, actual values cannot be determined for any time points. Ifu counts should be matched with genome counts for each temperature throughout the entire developmental cycle.*

qPCR from populations is inherently noisy and not easily compared to following gene expression of a single inclusion over time, which is why we developed this non destructive method to assess development in single inclusions. Any destructive population based method, especially late in infection, would not accurately capture the kinetics of individual inclusions and would not generate a useful comparison as some cells lyse and reinfection occurs confounding analysis by population based methods. We have added further experiments following inclusions from beginning to end and changed the IFU figure to show the entire cycle.

Figure 4: 'Fluorescent intensities were normalized by respective MOI':

- Readers may be confused with the way the figure is presented. The authors report no differences, yet because they standardized to MOI, their graph appears to show differences in relative fluorescence between MOI groups. It would be simpler to show relative fluorescence values without normalizing by MOI.

We believe the normalized data better reflects the biology we are exploring. Higher MOIs would result in multiplying the cell numbers by the MOI so normalization allows us to see kinetic effects that are different from the expected multiple from the MOI.

- Absent normalization, this data would appear to indicate that all inclusions maintained the same eventual fluorescence level, regardless of MOI used in the starting inoculum. This is problematic, as after 18-20 hpi the fusion of inclusions should result in substantially more bacteria present per inclusion in high MOI infections than present in cells with an MOI of 1. In order to maintain equivalent mean fluorescence intensities, each bacterium within these fused inclusions would subsequently have to express less signal in order for the mean intensities to match those of the lower MOI infections. How do the author's account for this?

The vast majority of the inclusions cluster at the MTOC of the cell as soon as 4 hours post infection so they are measured together before and after fusion. We have made this clearer in the results. Our ROI captured all these clustered inclusions.

*- Rather than focusing only on the initiation of promoter activity, the authors should address the similarity in the maximal threshold fluorescence values between their test groups. While the onset of promoter activity may not deviate between MOI groups, one would expect that if more bacteria are present within a cell, then they should eventually generate MORE fluorescence signal, even if they initiate their promoter activates at the same time. That doesn't appear to be the case in the data provided. The obvious question is 'why?' Are promoters less active in each bacterium as a result of higher MOI infections? Is the assay approaching its upper detection limit and thus cannot account for higher intensity values associated with high MOI infections? This could potentially be the case for *hctAprom* and *scc2prom*, but it doesn't appear to be so for *ihcAprom* and *hctBprom*. Another important reason why validation of the limits of detection for this assay is critical for data interpretation.*

Our birth to death late data sheds more light on this. The MOI data was normalized to MOI as otherwise fluorescent would be a multiple of MOI. Our analysis of late time points gets difficult for high moi as cell lysis and superinfections become more common. Our interpretation of why they do not perfectly normalize as there is intracellular competition for a limiting nutrient so during linear EB production RBes divide more slowly and produce fewer EBs over time. But as there are more RBes the system looks to be going faster but upon normalization the system is slower per/RB

- Line 565: 'MOI does not affect RB to EB conversion'

Numerous studies have shown that fusogenic inclusions result in enhancement of the progression of Chlamydia's developmental cycle. At best, the data presented here suggests that MOI does not appear to affect 'the initiation of the RB to EB conversion' or 'does not affect RB to EB conversion early in the developmental cycle' or 'does not affect the first instances of RB to EB conversion.' As previously stated, nutrient limitation is unlikely to play any factor here, given that the experimental conditions are unlikely to significantly limit nutrients until later in the chlamydial developmental cycle.

That is what our data show.

- Figure 4D, Line 191-193: 'The lack of MOI response for the expression of EB genes corresponds closely with EB production as measured by a reinfection assay (Fig. 4E).'

o The authors need to state the limit of detection of their IFU / EB detection assay. Why are IFU counts only taken out to 24 hpi when they appear to be starting to diverge? Particularly as IFU counts are at close to the assay's limit of detection at early time points.

IFU calculations from populations are inherently noisy and not easily compared to following gene expression of a single inclusion over time, which is why we developed this non destructive method to assess development in single inclusions. Any destructive population based method, especially late in infection, would not accurately capture the kinetics of individual inclusions and would not generate a useful comparison as some cells lyse and reinfection occurs confounding analysis by population based methods. We have added further experiments following inclusions from beginning to end.

Figure 5:

- Fig 5F, line 219-221: 'We verified that superinfection had no effect on the production of infectious progeny by performing a replating assay in the presence of spectinomycin (Fig. 5F)'

o Again, how can the authors conclude that superinfection had no effect on EB production when they only look up until the 24 hpi time point? This time point coincidentally is where their assay is the least accurate, ie. close

to its limit of detection and prior to when nutrient limitation would likely begin to be a factor. There is an obvious divergence present at 24 hpi that likely would be significant at later time points.

IFU calculations from populations are inherently noisy and not easily compared to following gene expression of a single inclusion over time, which is why we developed this non destructive method to assess development in single inclusions. Any destructive population based method, especially late in infection, would not accurately capture the kinetics of individual inclusions and would not generate a useful comparison as some cells lyse and reinfection occurs confounding analysis by population based methods. We have added further experiments following inclusions from beginning to end.

Figure 6:

- Fig 6E: It is notable that IFU data post 24 hours is presented for penicillin-treatment experiments. It should be presented for previous experiments in Figure 4 and 5 as well.

Our data focuses on individual inclusions and IFU data was used to demonstrate that our assays act as a reasonable surrogate for IFU production. Focusing just on IFU data masks the underlying biology of chlamydial development as this data is an average of thousands of infected cells. IFU calculations from populations are inherently noisy and not easily compared to following gene expression of a single inclusion over time, which is why we developed this non destructive method to assess development in single inclusions. Any destructive population based method, especially late in infection, would not accurately capture the kinetics of individual inclusions and would not generate a useful comparison as some cells lyse and reinfection occurs confounding analysis by population based methods. We have added further experiments following inclusions from beginning to end.

For the penicillin-treatment studies:

- Have the authors considered how the continuous replication of the promoter-reporter plasmid within penicillin-treated cells will affect their readouts?

Although we did not measure this we believe that plasmid replication and genomic replication will both respond to the pen treatment similarly. That is they both continue as does gene expression.

Fig. S3: "EB production follows a linear trajectory"

- This is not an accurate statement, as corresponding IFUs were not established. Rather hctB expression appears to follow a linear trajectory.

IFU calculations from populations are inherently noisy and not easily compared to following gene expression of a single inclusion over time, which is why we developed this non destructive method to assess development in single inclusions. Any destructive population based method, especially late in infection, would not accurately capture the kinetics of individual inclusions and would not generate a useful comparison as some cells lyse and reinfection occurs confounding analysis by population based methods. We have added further experiments following inclusions from beginning to end.

Minor comments:

Lines 146-147: "The initial expression from the ihtA promoter was in good agreement with the initiation of RB cell division as demonstrated by genome copy numbers .."

- an increase in genome copy suggests that cell division has occurred, but this is not definitive. DNA replication is only one step in the cell division process. This is an important distinction, as there is an abundance of data

showing that the 2 processes are decoupled in bacteria in general, and in Chlamydia in particular, with septation often lagging substantially behind DNA replication.

Although there is some truth to this, the general lag is almost always less than 2 genomes under normal growth conditions. Replication indexes in rapidly growing bacteria can hit 1.6 or so. We expect Chlamydia under normal growth conditions to behave similarly, supporting our use of genome copy numbers as an indicator of the initiation of replication.

*Line 148: "The initiation of RB replication signals the end of the EB to RB transition after cell entry."
- this is true for individual bacteria but the EB to RB conversion is not always synchronous across all cells in an infected monolayer, even when cells are infected by centrifugation.*

In our experiments initiation of gene expression demonstrated a high level of synchrony, our videos support this. We also follow single inclusion traces and average them therefore if there are shifts in individuals then there would be shifts in the population. We would expect the differences in synchronization to be normally distributed within and between samples since they are internally controlled.

Line236: There is a considerable amount of published literature linking cell division to EB development in Chlamydia species. These studies should be referenced in the introduction or leading up to these experiments.

We have added any relevant references

Line 262: It is important to note that C. trachomatis does not encode an alanine racemase.

We have changed the wording of this.

Line 281-283: This is unsurprising, given that IFN-induced persistence is thought to function by significantly restricting protein synthesis by the microbe. Rather than indicating a loss of promoter activity, this result is likely due to the effect of tryptophan-depletion on general translation under your treatment conditions.

Yes we also interpret the lack of any gene expression as a general lack of translation. We have made this clearer in the discussion.

Line 308-309: The authors might consider how maintaining an increasing number of metabolically active EBs within an inclusion might affect intracellular nutrient availability over time.

We have added additional data focused on the late stages of the infection and discussion on the interpretation of our late data analysis.

Line 344-346: If this interpretation is correct, then the authors should expect total EB numbers achievable between the 37 and 40C groups to be equivalent. Alternatively, if 40C is simply an inhospitable temperature for Chlamydial development, EB production may simply initiate early, and result in lower infectious progeny production.

We have added additional data focused on the late stages of the infection and included the IFU for the entire cycle.

Line 367-369: This is description of the aberrance phenotype. 'Persistence' indicates that this phenotype is reversible.

The current literature does not always differentiate between aberrant forms and persistent forms as few studies test reversibility but as we also did not assess reversibility will we use the more descriptive term “aberrant forms” in the manuscript.

Line 373-377: As previously indicated, because the assay is dependent on protein translation for its readout, it is not an accurate way to measure gene expression when protein synthesis is likely compromised, as is the case with IFN-induced persistence.

We disagree and believe that protein production is the best readout for assessing gene expression. It is generally agreed upon that unless the product is a regulatory RNA species a gene is not expressed until the protein is expressed.

Line 431-433: It is important to note that these studies indicate that polarized division occurs not just at the mid-late stages of development, but throughout the entirety of the developmental cycle.

Yes, our current model proposes 3-5 RB_R divisions followed by asymmetrical RB_E divisions

August 24, 2020

Dr. Scott S Grieshaber
University of Idaho
Biological Sciences
Moscow, ID 83844

Re: mSystems00689-20 (Single Inclusion Kinetics of Chlamydia trachomatis Development)

Dear Dr. Scott S Grieshaber:

Thank you for your resubmission of this manuscript which has now been evaluated by two reviewers. Please read through their reviews which should be addressed before your paper can be accepted.

In particular, Reviewer 2 has raised a substantive issue regarding the possibility that translation is affected globally under some conditions and that fluorescent protein production cannot be unambiguously assigned to transcriptional control. I recommend either incorporating additional experimental data, or carefully rewriting the discussion to lay out the logical possibilities.

Below you will find the comments of the reviewers.

To submit your modified manuscript, log onto the eJP submission site at <https://msystems.msubmit.net/cgi-bin/main.plex>. If you cannot remember your password, click the "Can't remember your password?" link and follow the instructions on the screen. Go to Author Tasks and click the appropriate manuscript title to begin the resubmission process. The information that you entered when you first submitted the paper will be displayed. Please update the information as necessary. Provide (1) point-by-point responses to the issues raised by the reviewers as file type "Response to Reviewers," not in your cover letter, and (2) a PDF file that indicates the changes from the original submission (by highlighting or underlining the changes) as file type "Marked Up Manuscript - For Review Only."

Due to the SARS-CoV-2 pandemic, our typical 60 day deadline for revisions will not be applied. I hope that you will be able to submit a revised manuscript soon, but want to reassure you that the journal will be flexible in terms of timing, particularly if experimental revisions are needed. When you are ready to resubmit, please know that our staff and Editors are working remotely and handling submissions without delay. If you do not wish to modify the manuscript and prefer to submit it to another journal, please notify me of your decision immediately so that the manuscript may be formally withdrawn from consideration by mSystems.

To avoid unnecessary delay in publication should your modified manuscript be accepted, it is important that all elements you upload meet the technical requirements for production. I strongly recommend that you check your digital images using the Rapid Inspector tool at <http://rapidinspector.cadmus.com/RapidInspector/zmw/>.

If your manuscript is accepted for publication, you will be contacted separately about payment when the proofs are issued; please follow the instructions in that e-mail. Arrangements for payment

must be made before your article is published. For a complete list of **Publication Fees**, including supplemental material costs, please visit our website.

Sincerely,

Michael Rust

Editor, mSystems

Journals Department
Reviewer comments:

Reviewer #1 (Comments for the Author):

The authors work to carefully address the comments by reviewers, which in some cases, were awfully detailed. The manuscript is very interesting and works to develop a novel computer-based and microscopy-based system to investigate chlamydial development within cells. The approach is quite novel; to my knowledge, unprecedented in the chlamydial research community. The methodology is well developed and could be repeated by individuals with the proper equipment and appropriate recombinant chlamydial strains. The work presents a solid quantitative and data-driven base for individuals who might wish to explore different stimuli or growth conditions in their study of chlamydial intracellular development. These strengths lead to a conclusion that the work is very important and will be interesting to a wide variety of readers.

The following is a list of modest concern that should be addressed by the authors.

Check the italics for Chlamydia throughout: make sure they are used consistently with your purpose.

There are lots of acronyms- perhaps a few could be removed for clarity. One would be "Genome Copies". It is used a lot, but I think it could be spelled out throughout.

Line 225 and elsewhere: make certain that the distinction between host cell division and bacterial cell division is clear.

Line 295: Euoprom should be written as is done elsewhere in the text.

The references on lines 313 and 315 are not structured correctly.

Line 365: fusogenic is spelled wrong.

Lines 388 and 407, and following text: These concepts should be presented together.

Reviewer #2 (Comments for the Author):

The authors have made a concerted effort to clarify a number of points in their introduction, experimental design and discussion sections, as well as justifying their approaches and acknowledging the limits of some of those approaches. This is a difficult system to work in, and the authors' have done a good job addressing the majority of concerns with their data late in the developmental cycle. The addition of the data from later time points in the developmental cycle and the comparison of infected cells treated with cycloheximide are great additions, and significantly strengthen the manuscript. Only one major criticism has not been adequately addressed and I have only a few minor additional suggestions:

Major (unaddressed) comment:

Author's response: As we analyzed individual inclusions under a variety of conditions that affect growth rate (temperature, super infection, MOI, cycloheximide), we do not agree that variations in protein synthesis would impact our interpretation of the data. The reporters all acted as would be expected across these conditions.

Line 414-419: Comparing the live-cell imaging data from these different persistence inducers revealed that the IFN γ treated Chlamydia never expressed Clover from any promoters tested early or late. This was also true for Chlamydia grown in the presence of the iron chelator bipyridyl.

Like all reporter assays, the one used in this manuscript is dependent on 1) the transcription and 2) the translation of the fluorescent product. Anything that globally affects 1) transcription or 2) translation will affect the reporters used in this assay. The proof is in the interferon and iron-limitation experiments, where the author's show an absence of any fluorescent activity for all of their promoter fusion constructs. Both interferon and iron-sequestration result in a global downregulation in translation in *C. trachomatis*, which has been well documented, and this is the most likely explanation for the author's results. Rather than showing that Clover is never transcribed, the authors only show that Clover is not translated. A confirmation of the presence of the transcripts for each construct being transcribed would show whether this block on Clover expression was at the level of RNA or protein. The concern is that other experimental conditions tested similarly affect protein translation in general, rather than the transcription of specific genes controlled by the various promoters under investigation.

As stated previously, this is an inherent limitation of all reporter fusion systems, which is why controls are generally used to eliminate the possibility that differences in promoter activity in an experiment are due to global effects on transcription / translation, rather than specific to a target promoter of interest. That, or changes in RNA transcripts are measured in tandem to determine whether they align with changes in fluorescent activity. If the authors do not wish to conduct these

controls, then they need to more clearly state this inherent limitation in their results or discussion sections.

Minor comments:

Line 156-157: Referencing other studies showing temperature enhancing growth rate in *C. trachomatis* and/or other *Chlamydia* species and any known mechanisms by which this occurs (enhanced metabolism?) would be beneficial.

Line 318-320: Given how cycloheximide is thought to affect *Chlamydia* development and the difference in the initiation timing and the slopes between treated and untreated for euoprom, I don't see how the authors could conclude that "... EB production ... is not likely a response to changing environmental signals (Line 326-327). If the growth rate of *Chlamydia* per cell is limited by steady state levels of a limiting nutrient provided by the host, then it would seem that this nutrient is not present at steady state levels until after 20 hpi in untreated cells, but is present in treated cells. The euoprom actually looks more linear under treated conditions: can the authors compare the Δ slope between the euoprom and hctBprom and report this as a comparable ratio for treated and untreated conditions?

Line 454-463: The authors might also consider that a reduction in cell size over time would drastically enhance the surface area to volume ratio of each chlamydial cell. This would likely impact localized concentrations of a number of metabolites and the rate at which they are imported / exported across the bacterial cell membrane.

Reviewer comments:

Reviewer #1 (Comments for the Author):

The authors work to carefully address the comments by reviewers, which in some cases, were awfully detailed. The manuscript is very interesting and works to develop a novel computer-based and microscopy-based system to investigate chlamydial development within cells. The approach is quite novel; to my knowledge, unprecedented in the chlamydial research community. The methodology is well developed and could be repeated by individuals with the proper equipment and appropriate recombinant chlamydial strains. The work presents a solid quantitative and data-driven base for individuals who might wish to explore different stimuli or growth conditions in their study of chlamydial intracellular development. These strengths lead to a conclusion that the work is very important and will be interesting to a wide variety of readers.

The following is a list of modest concerns that should be addressed by the authors.

Check the italics for Chlamydia throughout: make sure they are used consistently with your purpose.

Fixed

There are lots of acronyms- perhaps a few could be removed for clarity. One would be "Genome Copies". It is used a lot, but I think it could be spelled out throughout.

Fixed

Line 225 and elsewhere: make certain that the distinction between host cell division and bacterial cell division is clear.

Fixed

Line 295: Euoprom should be written as is done elsewhere in the text.

Fixed

The references on lines 313 and 315 are not structured correctly.

Fixed

Line 365: fusogenic is spelled wrong.

Fixed

Lines 388 and 407, and following text: These concepts should be presented together.

We prefer to keep these concepts separate as we are highlighting Pen and DCS's effects on cell division and not necessarily aberrancy/persistence. We include the IFN γ and bipyridyl results just for comparison.

Reviewer #2 (Comments for the Author):

The authors have made a concerted effort to clarify a number of points in their introduction, experimental design and discussion sections, as well as justifying their approaches and acknowledging the limits of some of those approaches. This is a difficult system to work in, and the authors' have done a good job addressing the majority of concerns with their data late in the developmental cycle. The addition of the data from later time points in the developmental cycle and the comparison of infected cells treated with cycloheximide are great additions, and

significantly strengthen the manuscript. Only one major criticism has not been adequately addressed and I have only a few minor additional suggestions:

Major (unaddressed) comment:

Author's response: As we analyzed individual inclusions under a variety of conditions that affect growth rate (temperature, super infection, MOI, cycloheximide), we do not agree that variations in protein synthesis would impact our interpretation of the data. The reporters all acted as would be expected across these conditions.

Line 414-419: Comparing the live-cell imaging data from these different persistence inducers revealed that the IFN γ treated Chlamydia never expressed Clover from any promoters tested early or late. This was also true for Chlamydia grown in the presence of the iron chelator bipyridyl.

*all reporter assays, the one used in this manuscript is dependent on 1) the transcription and 2) the translation of the fluorescent product. Anything that globally affects 1) transcription or 2) translation will affect the reporters used in this assay. The proof is in the interferon and iron-limitation experiments, where the authors show an absence of any fluorescent activity for all of their promoter fusion constructs. Both interferon and iron-sequestration result in a global downregulation in translation in *C. trachomatis*, which has been well documented, and this is the most likely explanation for the author's results. Rather than showing that Clover is never transcribed, the authors only show that Clover is not translated. A confirmation of the presence of the transcripts for each construct being transcribed would show whether this block on Clover expression was at the level of RNA or protein. The concern is that other experimental conditions tested similarly affect protein translation in general, rather than the transcription of specific genes controlled by the various promoters under investigation.*

As stated previously, this is an inherent limitation of all reporter fusion systems, which is why controls are generally used to eliminate the possibility that differences in promoter activity in an experiment are due to global effects on transcription / translation, rather than specific to a target promoter of interest. That, or changes in RNA transcripts are measured in tandem to determine whether they align with changes in fluorescent activity. If the authors do not wish to conduct these controls, then they need to more clearly state this inherent limitation in their results or discussion sections.

We have made no claims about the underlying mechanisms regulating gene expression. There are likely global, metabolism linked effects as well as specific cell type regulation effects. These also likely operate at the transcriptional and translational level. However, we are reporting the outcome of these regulatory mechanisms and we strongly believe that protein production is the best readout for assessing gene expression. It is generally agreed upon that unless the product is a regulatory RNA species a gene is not expressed until the protein is produced.

Minor comments:

*Line 156-157: Referencing other studies showing temperature enhancing growth rate in *C. trachomatis* and/or other Chlamydia species and any known mechanisms by which this occurs (enhanced metabolism?) would be beneficial.*

We have added an additional reference (Sturm, A. W. et al. 2011, November 1. Differences in Chlamydia trachomatis growth rates in human keratinocytes among lymphogranuloma venereum reference strains and clinical isolates. *Journal of Medical Microbiology*.)

Line 318-320: Given how cycloheximide is thought to affect Chlamydia development and the difference in the initiation timing and the slopes between treated and untreated for euoprom, I don't see how the authors could conclude that "... EB production ... is not likely a response to changing environmental signals (Line 326-327). If the growth rate of Chlamydia per cell is limited by steady state levels of a limiting nutrient provided by the host, then it would seem that this nutrient is not present at steady state levels until after 20 hpi in untreated cells, but is present in treated cells. The euoprom actually looks more linear under treated conditions: can the authors compare the Δ slope between the euoprom and hctBprom and report this as a comparable ratio for treated and untreated conditions?

Euo expression follows a logistical curve so a slope cannot be measured. However, time to half max of EUO is sooner and HctB slope is higher in the cycloheximide treated infections which is consistent with our interpretation that growth rate and cell division are linked to development.

Line 454-463: The authors might also consider that a reduction in cell size over time would drastically enhance the surface area to volume ratio of each chlamydial cell. This would likely impact localized concentrations of a number of metabolites and the rate at which they are imported / exported across the bacterial cell membrane.

Yes, this could contribute to the early switch between RB to RB division and RB to IB division. But we did not investigate any mechanism for this switch.

September 11, 2020

Dr. Scott S Grieshaber
University of Idaho
Biological Sciences
Moscow, ID 83844

Re: mSystems00689-20R1 (Single Inclusion Kinetics of Chlamydia trachomatis Development)

Dear Dr. Scott S Grieshaber:

As I understand it, the issue raised by reviewer 2 is not to dispute the idea that the translated protein is the ultimate product of gene expression. Rather, through out the manuscript, including in the abstract, there are mentions of results on "promoter activity" and "promoter reporters". Strictly, these fluorescent are convolving the transcriptional activity of the promoter with translation and maturation of the fluorophore. Without measuring transcripts directly I think it is appropriate when revising the manuscript to pay careful attention to this language and include a caveat in the discussion that there may be post-transcriptional control that contributes to the results.

Below you will find the comments of the reviewers.

To submit your modified manuscript, log onto the eJP submission site at <https://msystems.msubmit.net/cgi-bin/main.plex>. If you cannot remember your password, click the "Can't remember your password?" link and follow the instructions on the screen. Go to Author Tasks and click the appropriate manuscript title to begin the resubmission process. The information that you entered when you first submitted the paper will be displayed. Please update the information as necessary. Provide (1) point-by-point responses to the issues raised by the reviewers as file type "Response to Reviewers," not in your cover letter, and (2) a PDF file that indicates the changes from the original submission (by highlighting or underlining the changes) as file type "Marked Up Manuscript - For Review Only."

Due to the SARS-CoV-2 pandemic, our typical 60 day deadline for revisions will not be applied. I hope that you will be able to submit a revised manuscript soon, but want to reassure you that the journal will be flexible in terms of timing, particularly if experimental revisions are needed. When you are ready to resubmit, please know that our staff and Editors are working remotely and handling submissions without delay. If you do not wish to modify the manuscript and prefer to submit it to another journal, please notify me of your decision immediately so that the manuscript may be formally withdrawn from consideration by mSystems.

Sincerely,

Michael Rust

Editor, mSystems

Journals Department
Reviewer comments:

September 15, 2020

Dr. Scott S Grieshaber
University of Idaho
Biological Sciences
Moscow, ID 83844

Re: mSystems00689-20R2 (Single Inclusion Kinetics of Chlamydia trachomatis Development)

Dear Dr. Scott S Grieshaber:

Your manuscript has been accepted, and I am forwarding it to the ASM Journals Department for publication. For your reference, ASM Journals' address is given below. Before it can be scheduled for publication, your manuscript will be checked by the mSystems senior production editor, Ellie Ghatineh, to make sure that all elements meet the technical requirements for publication. She will contact you if anything needs to be revised before copyediting and production can begin. Otherwise, you will be notified when your proofs are ready to be viewed.

Sincerely,

Michael Rust
Editor, mSystems

Journals Department
Fig. S3: Accept

ST1: Accept

Supplemental Material: Accept

Fig. S1: Accept

Fig. S4: Accept

Fig. S2: Accept

Supplemental Material: Accept

Fig. S5: Accept